# The Copernicus Polar Ice and Snow Topography Altimeter (CRISTAL) High-Priority Candidate Mission

Michael Kern[1], Robert Cullen[1], Bruno Berruti[1], Jerome Bouffard[2], Tania Casal[1], Mark R. Drinkwater[1], Antonio Gabriele[1], Arnaud Lecuyot[1], Michael Ludwig[1], Rolv Midthassel[1], Ignacio Navas Traver[1], Tommaso Parrinello[2], Gerhard Ressler[1], Erik Andersson[3], Cristina Martin-Puig[4], Ole Andersen[5], Annett Bartsch[6], Sinead Farrell[7], Sara Fleury[8], Simon Gascoin[9], Amandine Guillot[10], Angelika Humbert[11], Eero Rinne[12], Andrew Shepherd[13], Michiel R. van den Broeke[14], John Yackel[15]

[1]European Space Agency (ESA-ESTEC), Keplerlaan 1, 2201 AZ Noordwijk, The Netherlands
[2]European Space Agency (ESA-ESRIN), Via Galileo Galilei, Casella Postale 64, 00044 Frascati, Italy
[3]European Commission, BREY 09/154, 1049 Brussels, Belgium
[4]EUMETSAT, Eumetsat Allee 1, 64295 Darmstadt, Germany
[5]DTU Space, Elektrovej 28, 2800 Lyngby, Denmark
[6]b.geos, Industriestrasse 1, 2100 Korneuburg, Austria
[7]University of Maryland, 5825 University Research Court, 20740 College Park, MD, USA
[8]CTOH/LEGOS/CNRS, 14 avenue Edouard Belin, 31400 Toulouse, France
[9]CESBIO, Université de Toulouse, CNRS/CNES/IRD/UPS, 31400 Toulouse, France
[10]CNES, 18 avenue Edouard Belin, 31400 Toulouse, France
[11]Alfred-Wegner-Institute Helmholtz Centre for Polar and Marine Research, Am Alten Hafen 26, 2758 Bremerhaven, Germany
[12]Finnish Meteorological Institute, PO Box 503, 00101 Helsinki, Finland
[13]Centre for Polar Observation and Modelling, University of Leeds, LS2 9JT, UK
[14]Utrecht University, Princetonplein 5, 3584 CC Utrecht, The Netherlands
[15]University of Calgary, 2500 University Drive NW, Earth Sciences 356 Calgary, Alberta, Canada

*Correspondence to*: M. Kern (Michael.Kern@esa.int)

**Abstract.**

The Copernicus polaR Ice and Snow Topography ALtimeter (CRISTAL) is one of six high-priority candidate missions under consideration by the European Commission to enlarge the Copernicus Space Component. Together, the high-priority candidate missions fill gaps in the measurement capability of the existing Copernicus Space Component to address emerging and urgent user requirements in relation to monitoring anthropogenic $CO_2$ emissions, polar environments, and land surfaces. The ambition is to enlarge the Copernicus Space Component with the high-priority candidate missions in the mid-2020s to provide enhanced continuity of services in synergy with the next generation of the existing Copernicus Sentinel missions. CRISTAL will carry a dual-frequency synthetic aperture radar altimeter as its primary payload for measuring surface height, and a passive microwave radiometer to support atmospheric corrections and surface-type classification. The altimeter will have interferometric capability at Ku-band for improved ground resolution, and a second (non-interferometric) Ka-band frequency to provide information on snow layer properties. This paper outlines the user consultations that have supported expansion of the Copernicus Space Component to include the high-priority candidate missions, describes the primary and secondary

objectives of the CRISTAL mission, identifies the key contributions the CRISTAL mission will make, and presents a concept - as far as it is already defined - for the mission payload.

## 1 Introduction

Earth's cryosphere plays a critical role in our planet's radiation and sea level budgets. Loss of Arctic sea ice is exacerbating planetary warming owing to the ice-albedo feedback (e.g. Budyko, 1969; Serreze and Francis, 2006; Screen and Simmonds 2010), and loss of land ice is the principal source of global sea level rise, see (Intergovernmental Panel on Climate Change (IPCC)/ SROCC, 2019). The rates and magnitudes of depletion of Earth's marine and terrestrial ice fields are among the most significant elements of future climate projections (Meredith et al, 2019). The Arctic provides fundamental ecosystem services (including fisheries management and other resources), sustains numerous indigenous communities, and due to sea ice loss is emerging as a key area for economic exploitation. The fragile ecosystems are subject to pressures from a growing number of maritime and commercial activities. The potentially-devastating contribution of the Antarctic ice sheet to global sea level rise is also subject to large uncertainties in ice mass loss, with high-end estimates of sea-level contribution exceeding a metre of global mean sea-level rise by 2100 (Edwards et al., 2019).

A long-term programme to monitor the Earth's polar ice, ocean and snow topography is important to stakeholders with interests in the Arctic and Antarctic. While Europe has a direct interest in the Arctic due to its proximity (see https://ec.europa.eu/environment/efe/news/integrated-eu-policy-arctic-2016-12-08_en), the Arctic is also of interest to other countries and international communities. Changes in the Arctic environment affect strategic areas including politics, economics (e.g. exploitation of natural resources including minerals, oil and gas, fish) and security. Besides economic impacts of Antarctic and Arctic changes (Whiteman et al, 2013), Europe's interest in both polar regions is due to their influence on patterns and variability in global climate change, thermohaline circulation and the planetary energy balance. Last but not least, changes in the Arctic system have potential impacts on weather, with consequences for extreme events (Francis et al., 2017). The Copernicus polaR Ice and Snow Topography Altimeter (CRISTAL) mission, described in this paper, addresses the data and information requirements of these user communities with a particular focus on addressing Copernicus service requirements.

In the following section, we provide a background of the Copernicus programme and candidate missions that are being prepared by the European Space Agency (ESA) in partnership with the European Union (EU) in response to Copernicus user needs. In Section 3, we describe the objectives of the CRISTAL mission and its relation to the Copernicus services. We then discuss the key contributions from the CRISTAL mission, both in terms of specific mission objectives as well as expected scientific contributions towards improved knowledge in Section 4. In Section 5, an overview of CRISTAL's current system concept and mode of operation is described. This section also highlights the use of heritage technology and needs driving

technical advancements to improve observational capabilities beyond current missions. Conclusions and a current mission status statement are provided in Section 6.

## 2 Expansion and evolution of the Copernicus Space Component

Copernicus was established to fulfil the growing need amongst European policymakers to access accurate and timely information services to better manage the environment, understand and mitigate the effects of climate change and ensure civil security. To ensure the operational provision of Earth-observation data, the Copernicus Space Component (CSC) includes a series of seven space missions called 'Copernicus Sentinels', which are being developed by ESA specifically for Global Monitoring for Environment and Security (GMES)/Copernicus. The Copernicus programme is coordinated and managed by the European Commission (EC). It includes Earth observation satellites, ground-based measurements, and services to process data to provide users with reliable and up-to-date information through a set of Copernicus Services related to environmental and security issues.

The intense use of Copernicus has generated high expectations for an evolved Copernicus system. There is now a large set of defined needs and requirements. With respect to the polar regions, user and observation requirements have been identified, structured and prioritised in a process led by the EC (Duchossois et al., 2018a; 2018b). Two distinct sets of expectations have emerged from this user consultation process. Firstly, stability and continuity, while increasing the quantity and quality of Copernicus products and services, led to one set of requirements. They are distinctly addressed in the considerations for the next generation of the current Sentinels 1 to 6 series, see e.g. European Commission (2017). Emerging and urgent needs for new types of observations constitute a second distinct set of requirements that are mainly addressed through evolution of the Copernicus Space Segment service. This evolution corresponds to the enlargement of the present space-based measurement capabilities through the introduction of new missions to answer these emerging and urgent user requirements. After extensive consultation, six potential high-priority candidate missions have been identified (ESA, 2019b); the Copernicus Hyperspectral Imaging Mission for the Environment (CHIME), the Copernicus Imaging Microwave Radiometer (CIMR), the Copernicus Anthropogenic Carbon Dioxide Monitoring mission (CO2M), the Copernicus Polar Ice and Snow Topography Altimeter (CRISTAL), the Copernicus Land Surface Temperature Monitoring mission (LSTM), and the L-band Synthetic Aperture Radar (ROSE-L).

## 3 Objectives of the CRISTAL mission

The strategic, environmental and socio-economic importance of the Arctic region has been emphasised by the European Union in their integrated policy for the Arctic (https://ec.europa.eu/environment/efe/news/integrated-eu-policy-arctic-2016-12-08_en), including the Arctic Ocean and its adjacent seas. Considering the sparse population and the lack of transport links, a

capacity for continuous monitoring the Arctic environment with satellites is considered essential. In light of this, and of the importance of the polar regions more widely, guiding documents have been prepared in a European Commission-led user consultation process: the Polar Expert Group (PEG) User Requirements for a Copernicus Polar Mission Phase-I report, Duchossois et al. (2018a), hereafter referred to as PEG-1 report, and the Phase 2 report on Users' requirements, Duchossois et al. (2018b), hereafter referred to as PEG-2 report.

The required geophysical parameters for the polar regions are summarised and prioritised in the PEG-1 report, which addresses objectives as defined in the EU Artic Policy Communication, namely: climate change, environmental safeguarding, sustainable development, support to indigenous populations and local communities. Floating ice parameters were listed as the top priority for a polar mission considering the availability of existing Copernicus products and services, their needs for improvement (e.g. in terms of spatial resolution, and accuracy) and the current level of their technical and/or scientific maturity. The specific parameters include sea ice extent, concentration, thickness, type, drift, and velocity, thin ice distribution, iceberg detection, drift and volume change as well as ice shelf (the floating extension of the ice sheets) thickness and extent. These parameters were given a top priority by the European Commission due to their key position in operational services such as navigation and marine operations, meteorological and seasonal prediction, and climate model validation. The PEG-1 report also stresses the importance of a measuring capability for mountain glaciers and ice caps, seasonal snow, ice sheets, oceans, freshwater, and permafrost.

The Global Climate Observing System (GCOS, 2011) have stated that actions should be taken to ensure continuation of altimeter missions over sea ice. They suggested continuation of satellite Synthetic Aperture Radar (SAR) altimeter missions, with enhanced techniques for monitoring sea ice thickness, to achieve capabilities to produce time series of monthly, 25 km sea ice thickness with 0.1 m accuracy for polar regions. It was mentioned that near-coincident data would help resolve uncertainties in sea ice thickness retrieval. Such measurements could be achieved, for example, through close coordination between radar and laser altimeter missions. In addition to sea ice thickness, other sea ice parameters retrievable from SAR altimetry, such as ice drift, shear and deformation, leads and ice ridging, were pointed to as observables for future improvement.

While the Copernicus Sentinel-3 mission provides partial altimetric measurements of the polar oceans, their inclination limits the coverage to latitudes between 81.5°N and 81.5°S. With the expected on-going loss of Arctic sea ice, these satellites will monitor only a small amount of the Arctic ice cover during summer periods by mid-2020, see e.g. Quartly et al (2019). Currently, ESA's CryoSat-2 (Drinkwater et al., 2004; Wingham et al., 2006, Parrinello et al. 2018) is the only European satellite to provide monitoring of the oldest, thickest multiyear ice. However, continued monitoring of the polar regions – and the Arctic Ocean north of 81.5°N in particular - are at risk, since CryoSat-2 has been operating in its extended mission scenario since its nominal end-of-mission lifetime of October 2013 (see Figure 1). This risk has widely been recognised by the polar and ocean surface topography community. For example, at the  Ocean Surface Topography Science Team (OSTST) 2019

meeting (Chicago, IL, USA, 21-25 October 2019) a recommendation was recorded (in view of the preparations for CRISTAL and other missions currently in operation): "To minimize likelihood of a gap in polar ocean and ice monitoring, the OSTST encourages Agencies to strive to launch a high-resolution polar altimeter in the early 2020s (such as the proposed HPCM CRISTAL) and to maintain operation of CryoSat-2, ICESat-2, and SARAL/AltiKa as long as possible".

Based on the user requirements and priorities outlined in the PEG-1 report, a set of high-priority mission parameters were defined by ESA's CRISTAL Mission Advisory Group (MAG) and ESA, which led to the CRISTAL mission objectives (Table 1). The primary objectives drive the design and performance specifications of the CRISTAL mission, whereas the secondary objectives reflect the opportunity to support a wider range of users and services.

**Table 1 CRISTAL mission objectives**

| Nature | Target | Objective |
|---|---|---|
| Primary | Sea ice | **To measure and monitor variability of Arctic and Southern Ocean sea ice thickness and its snow depth.** Seasonal sea ice cycles are important for both human activities and biological habitats. The seasonal to inter-annual variability of sea ice is a sensitive climate indicator; it is also essential for long term planning of any kind of activity in the polar regions. Knowledge of snow depth will lead to improved accuracy in measurements of sea ice thickness and is also a valuable input for coupled atmosphere-ice-ocean forecast models. On shorter timescales, measurements of sea ice thickness and information about Arctic Ocean sea state are essential support to maritime operations over polar oceans. |
| Primary | Land ice | **To measure and monitor the surface elevation and changes therein of polar glaciers, ice caps and the Antarctic and Greenland ice sheets.** The two ice sheets of Antarctica and Greenland store a significant proportion of global fresh water volume and are important for climate change and contributions to sea level. Monitoring grounding line migration and elevation changes of floating and grounded ice sheet margins is important to identify and track emerging instabilities. These instabilities can negatively impact the stability of the ice sheets, leading to ice mass loss and accelerated sea-level rise. |
| Secondary | Ocean | **To contribute to the observation of global ocean topography as a continuum up to the polar seas.** Polar altimetry will contribute to the observation system for global observation of mean sea level, mesoscale and sub-mesoscale currents, wind speed and significant wave height. Information from this mission serves as critical input to operational oceanography and marine forecasting services in the polar oceans. |

| Secondary | Inland water | **To support applications related to coastal and inland waters. Observation of water level at Arctic coasts, as well as rivers and lakes are key quantities in hydrological research.** Rivers and lakes not only supply freshwater for human use, including agriculture, but also maintain natural processes and ecosystems. The monitoring of global river discharge and its long-term trend contributes to the evaluation of global freshwater flux that is critical for understanding the mechanism of global climate change (Prowse et al, 2011), Zakharova et al, 2020). Changes to seasonal freezing of Arctic river and lakes, in the context of climate change, will also be important to study and understand. Their observation could help forecasting their evolutions and organizing alternative modes of transport. |
|-----------|------|-----|
| Secondary | Snow | **To support applications related to snow cover and permafrost in Arctic regions.** Snowmelt timing is a key parameter for hydrological research, since it modulates the river discharge of Arctic basins (Shiklomanov et al, 2007). Surface state change in permafrost regions indicates the initiation of ground thaw and soil microbial activities in the seasonally unfrozen upper soil (active) layer. The rapid evolution of the permafrost has also important impacts on human activities and infrastructures. |

By addressing these objectives, the mission responds to a number of required parameters of interests and applications in Copernicus Services. A mapping of the services to the parameters of interest and applications is listed in Table 2.

**Table 2 Copernicus Services addressed by CRISTAL.**

| Copernicus Service | Relevant geophysical parameters of interest | Core information service addressed or affected (forecasting, monitoring or projections) |
|---|---|---|
| Copernicus Marine Environmental Monitoring Service (CMEMS) | • Sea ice thickness and snow depth<br>• Sea-level anomaly and geostrophic ocean currents in polar oceans<br>• Significant wave height in polar oceans<br>• Global sea level<br>• Global sea surface wind and waves | Maritime safety, Coastal and Marine Environment, Marine Resources and Weather, Seasonal Forecasting and Climate activities |
| Copernicus Climate Change Service (C3S) | • Ice-sheet topography<br>• Sea ice thickness and volumes<br>• Global sea level<br>• Snow depth over sea ice | Observations, Climate reanalysis, seasonal forecasts and climate projections |
| Copernicus Land Monitoring Service (CLMS) | • Ice-sheet and glacier topography | Biophysical monitoring, Land cover & land use mapping, Thematic hotspot mapping, Reference data, Ground motion service |

| Copernicus Atmospheric Monitoring Service (CAMS) | • Snow depth on sea ice | Meteorology and climatology seasonal forecasts and climate projections |
|---|---|---|
| Copernicus Emergency Management Service (CEMS) | • Lakes and rivers level/stage | Flood awareness forecast, Emergency Management System Mapping |

## 4 Key contributions of the CRISTAL mission

The following sections describe the key contributions of the mission in more detail, including the key requirements that guide the implementation of the mission.

### 4.1 Sea ice freeboard and thickness

Sea ice plays a critical role in Earth's climate system since it provides a barrier between the ocean and atmosphere, restricting the transfer of heat between the two. Due to its high albedo, the presence of sea ice reduces the amount of solar energy absorbed

by the ocean. Arctic sea ice rejects brine during formation and fresh water during melting and it is therefore a driving force of the global thermohaline circulation as well as the stratification of the upper layer of the ocean. The sea ice provides a critical habitat for marine mammals and for biological activity (e.g. Tynan et al, 2009), and it is platform enabling subsistence hunting and travel for indigenous coastal communities.

The sea ice cover of the Arctic Ocean is waning rapidly. By 2019, the decline in September Arctic sea ice extent was about 13% per decade, relative to the 1981–2010 average, and the older, thicker, multi-year ice cover comprised ~ 20 % of the winter ice pack, compared to ~ 45 % in the 1980s (Perovich et al., 2017, IPCC/SROCC, 2019). In the Southern Ocean sea ice is undergoing regional changes, with a decline observed in the Amundsen and Bellingshausen Seas (Shepherd et al., 2018). These losses are having a profound impact on the climate, environment and ecosystems of both polar regions. Monitoring the polar

oceans is therefore of regional and global importance, and the long-term continuity of sea ice measurements is essential to extending both climate and operational data services.

Global warming, and its Arctic amplification, continue to contribute to the decrease of multi-year ice in the central Arctic Ocean (north of 81.5° N). It is therefore critical to obtain continuous-, pan-Arctic observations of sea ice thickness, extending

as close as possible to the North Pole. Continuous monitoring of Arctic Ocean sea ice conditions is necessary for safe navigation through ice-covered waters. When linked to previous measurements from Envisat, ICESat, CryoSat-2 and ICESat-2, the CRISTAL mission will deliver observations that will provide a long-term record of sea ice thickness variability and trends that are critical to support climate services. Since sea ice thickness is an essential climate variable (see GCOS, 2011), its continuous measurement is required to understand the Arctic system and how ice loss is impacting global climate.


Shipping in ice-covered Arctic waters has increased significantly in recent years and is expected to continue to do so over the coming decades (IPCC/ SROCC, 2019). In addition to traditional maritime operations and fishing in the high Arctic, several polar-class cruise liners are under construction. This means an increase in the need and scope of operational ice information services. A primary data source for national ice services is currently synthetic aperture radar (SAR) imagery, specifically data

acquired by Sentinel-1A and 1B, RADARSAT-2 and RADARSAT Constellation Mission. Thus, independent measurements of sea ice thickness distribution at reasonable latencies provided by CRISTAL will complement existing SAR measurements and benefit operational ice charting. Furthermore, observed sea ice thickness or freeboard distributions can be assimilated into sea ice models to generate ice forecasts needed ice navigation and offshore operations.

Historically, satellite observations had primarily been used to monitor ice extent until Laxon et al. (2003) produced the first Arctic-wide sea ice thickness estimates from ERS radar altimetry. Since then, various methods for converting the received signal to physical variables have been established (Giles et al., 2008a, Laxon et al., 2013; Kurtz et al., 2014; Ricker et al., 2014; Price et al., 2015; Tilling et al., 2018; Hendricks et al., 2018). The capability to obtain an estimate of sea ice freeboard and thickness, and converting it to estimates of ice volume, has enabled scientists to better understand the changing Arctic ice

cover. Most recently, sea ice freeboard has been estimated from both Ka- and Ku-band measurements (Armitage and Ridout, 2015; Guerreiro et al., 2016; Lawrence et al., 2018).

Most sea ice thickness products are currently provided on a 25 km grid (see e.g. Sallila,et al 2019 for an overview of different products currently available), which corresponds to the GCOS user requirements (GCOS, 2011), but do not meet the specified

accuracy requirements of 0.1 m. The residual, systematic uncertainty in sea ice thickness is estimated to be 0.56-0.61 m for ICESat (Connor et al., 2013) and it is 0.6 m for CryoSat-2 observations over first year ice and 1.2 m for those over multi-year ice (Ricker et al., 2014). The uncertainty in ice thickness derived from CryoSat-2 observations is driven mainly by the unknown penetration of the radar pulse into the snow layer as a result of variable snow properties (Nandan et al., 2017, Nandan et al., 2020), as well as the choice of retracker (Ricker et al., 2014). Reference is also made to Mallett et al. (2020), which finds that

assumptions concerning the time evolution of overlying snow density can lead to underestimates of sea ice thickness from radar altimetry.

While the focus of the Copernicus programme is on the Arctic, comprising all areas north of the southernmost tip of Greenland (~ 60° N), the parameters specified for polar regions should equally be provided for its southern counterpart the Antarctic, as

well as all non-polar snow- and ice-covered surfaces.

The requirements for CRISTAL are currently stated to provide sea ice freeboard with an accuracy of 0.03 m along orbit segments of less than or equal to 25 km during winter months and to provide meaningful freeboard measurements during

summer months. Winter months are months from October to April in the Northern hemisphere and from May to October in the Southern hemisphere. The system shall be capable of delivering sea ice thickness measurements with a vertical uncertainty less than 0.15 m along orbit segments $\leq 25$ km in winter months, and provide meaningful sea ice thickness estimates during summer months. The along-track resolution of sea ice thickness measurements shall be at least 80 m. The uncertainty requirement for sea ice thickness comes with a caveat, as the thickness uncertainty depends on the uncertainty of auxiliary products. In the case of CRISTAL, snow thickness will be measured by the system, but snow and ice densities will still have to be estimated by other means. In light of the current 0.2 m sea ice thickness uncertainty from CryoSat-2 data assessed by Tilling et al. (2018) for a gridded, monthly product and the anticipated improvement from the dual-altimetry technology, especially in the snow depth and propagation estimates, reaching a higher vertical uncertainty would seem reachable but requires further study. Currently, the retrieval accuracy of sea ice freeboard is limited by the range resolution of a radar altimeter. The large bandwidth of 500MHz is an important driver for the CRISTAL instrument concept generation. A bandwidth of 500MHz will improve the range resolution from 50 cm (as for CryoSat-2 with 320 MHz bandwidth) to ~ 30 cm for CRISTAL.  A radiometer will help in active/passive synergy to classify sea ice type, see e.g. Tran et al (2009) for further justification.

## 4.2 Snow depth over sea ice

An accurate estimate of snow depth over Arctic sea ice is needed for signal propagation speed correction to convert radar freeboard to sea ice freeboard as well as conversion of freeboard to sea ice thickness (Laxon et al., 2003, 2013). The penetration aspects of a dual-frequency snow depth retrieval algorithm over Antarctica are complex (Giles et al., 2008b; Shepherd et al., 2018) and are not further elaborated here. In addition to uncertainty reduction for ice thickness/freeboard computation, the variation of snow depth is a parameter that is highly relevant for both climate modelling, ice navigation and polar ocean research. The snow climatology of Warren et al., (1999) is still the single most used estimate of snow depth in sea ice thickness processing (Sallila et al., 2019). The uncertainty in the original Warren snow depth estimates are halved over first year ice (Kurtz and Farrell, 2011, Zhou et al., 2020), but snow still represents still the single most important contribution to uncertainty in the estimation of sea ice thickness and volume (Tilling et al., 2018). The studies of Lawrence et al. (2018) and Guerreiro et al (2016) show the possibility of using Ku- and Ka-bands in mitigating the snow depth uncertainty. Dual-frequency methods improve the ability to reduce and estimate the uncertainties related to snow depth and sea ice thickness retrieval. The modelling community is particularly interested in the uncertainty information according to the user requirement study in the PEG-1 report. Better abilities to estimate the related uncertainties improves prediction quality assessment of annual snowmelt over Arctic sea ice (Blockley and Peterson, 2018). The stratigraphy and electromagnetic properties of the snow layer contrast with that of the underlying ice can be exploited to retrieve information on the snow layer properties if contemporaneous measurements are acquired from multiple scattering horizons. For details see Giles et al (2007), which demonstrated the propagating uncertainties associated with snow depth and other geophysical parameters. A dual-frequency satellite altimeter, as proposed for the CRISTAL mission, will address this need. CRISTAL aims to provide an uncertainty in snow depth retrieval over sea ice of

less than or equal to 0.05m. The additional measurements in Ka-band, with a 500MHz bandwidth, support the discrimination between the ice and snow interfaces.

## 4.3 Ice sheets, glaciers and ice caps

Earth's land ice responds rapidly to global climate change. For example, melting of glaciers, ice caps, and ice sheets over recent decades has altered regional and local hydrological systems, and has impacted sea levels and patterns of global ocean circulation. The Antarctic and Greenland ice sheets are Earth's primary freshwater reservoirs and, due to their progressive imbalance, have made an accelerating contribution to global sea level rise during the satellite era (Shepherd et al. 2018; Shepherd et al. 2019). Glaciers outside of the ice sheets constituted nearly 1/3 of all sea level rise over the past 2 decades

(Gardner et al., 2013; Wouters et al., 2019) Although ice dynamical models have improved, future losses from the polar ice sheets remain the largest uncertainty in sea level projections. Due to their continental scale, remote location, and hostile climatic environment, satellite measurements are the only practical solution for spatially and temporally complete monitoring of the polar ice sheets.

Estimates of ice sheet surface elevation change provide a wealth of geophysical information. They are used as the basis for computing the mass balance and sea level contribution of both ice sheets of Greenland and Antarctica (McMillan et al., 2014, 2016; Shepherd et al., 2012), for identifying emerging signals of mass imbalance (Flament and Rémy, 2012; Wingham et al., 2009) and for determining the loci of rapid ice loss (Hurkmans et al., 2014; Sørensen et al., 2015). Through combination with regional climate and firn models of surface processes, surface elevation change can be used to isolate ice dynamical changes,

at the scale of individual glacier catchments (McMillan et al., 2016).

A unique and continuous record of elevation measurements is provided by radar altimeters, dating back to 1992. The maps are typically delivered in (1) high-resolution (5-10 km) rates of surface elevation change (for single or multiple missions, typically computed as a linear rate of change over a period of several years to decades), and (2) frequently (monthly-quarterly) sampled

time series of the cumulative change, averaged across individual glacier basins. In addition to being used to quantify rates of mass balance and sea level rise, they also have a range of other applications, such as detection of subglacial lake drainage (Siegert et al., 2016) investigations of the initiation and speed of inland propagation of dynamic imbalance (Konrad et al., 2017) that provide valuable information relating to the underlying physical processes that drive dynamical ice loss.

CRISTAL will extend the decade's long record of the generation of elevation measurements provided by radar altimeters. It will produce maps of ice surface elevation with an uncertainty of 2 m (the vertical accuracy threshold is 2 m, an absolute accuracy of 0.5 m can be assumed and a relative accuracy goal of 0.2m). The system shall be capable of delivering surface elevation with an along-track resolution of at least 100 m with a monthly temporal sampling. CRISTAL will be capable of tracking steep terrain with slopes less than 1.5° using its SARIn mode. High-resolution Swath processing over ice sheets (about

5 km wide) can reveal complex surface elevation changes, related to climate variability and ice dynamics, and subglacial geothermal and magmatic processes, see e.g. Foresta et al (2016). Elevation measurements of regions with smaller glaciers are often missing in CryoSat-2 data. Indeed, tracking algorithms are not efficient when rough terrain is encountered. Improvement in the tracking over glaciers is thus a key element in the instrument concept generation.

## 4.4 Sea level, coastal and inland water

Over the years and through constant improvement of the data quality, satellite altimetry has been used in a growing number of applications in Earth sciences. The altimeter measurements are helping us to understand and monitor the ocean: its topography, dynamics and variability at different scales. The need of satellite observations to study, understand and monitor the ocean is more than essential over polar areas, where in-situ data networks are very sparse, and where profound and dramatic changes occur. This has also been expressed and emphasised by CMEMS as "Ensuring continuity (with improvements) of the Cryosat-285 2 mission for sea level monitoring in polar regions" (CMEMS, 2017). "Reliable retrieval of sea level in the sea ice leads to reach the retrieval accuracy required to monitor climate change" is another CMEMS recommendation for polar and sea ice monitoring, see CMEMS (2017).

Actual data from the CMEMS catalogue does not allow a satisfactory sampling north of 81.5°N. It is of prime importance that 290 the CRISTAL orbit configuration allows measurement coverage of the central Arctic Ocean with an omission not exceeding 2° of latitudes around the poles. SLA over frozen seas can only be provided by measurements in the leads. CRISTAL will contribute to the observation system for global observation of mean sea level, (sub-)mesoscale currents, wind speed and significant wave height as a critical input to operational oceanography and marine forecasting services, as well as supporting sea ice thickness retrieval in the Arctic.


The high inclination orbit of CRISTAL associated with high-resolution SAR/SARIn bi-band altimetry measurements would extend considerably our monitoring capability over the polar oceans. The development of tailored processing algorithms should have not only to track the low-frequency sea level trend in presence of sea ice and to characterize ocean large scale and mesoscale variations over regions not covered by conventional ocean altimeters. Beyond the observations of ice elevation 300 variations, CRISTAL would offer the unique opportunity to improve our knowledge on the mutual Ocean-Cryosphere interactions over short and long-term time scales for both poles. The Southern Ocean circulation plays a key role in shaping the Antarctic cryosphere environment. First, it regulates sea ice production: as sea ice forms and reject brines into the ocean, the ocean destabilizes and warms submerged waters that reaches the ocean surface, limiting further ice production. Second, it impacts Antarctic ice sheet melt, when warm and salty ocean currents access the base of floating glaciers through bathymetric 305 troughs of the Antarctic continental shelf. These ocean currents melt the ice shelves from below, and are the main causes of the current decline of floating ice-shelves (Shepherd et al., 2019, Smith et al., 2020). Thus melting of ice shelves represents one of the largest uncertainty in the current prediction of global sea-level change (Edwards et al., 2019), creating major gap in

our ability to respond and adapt to future climate. Tightly linked with glacier melt, the polar shelf circulation and its interaction with largescale circulation also control the rate of bottom water production and deep ocean ventilation that impact the world's oceans on timescale ranging from decades to millennia. Therefore, with a designed operational lifetime of at least 7.5 years (including in-orbit commissioning), the observation from the same sensor of each components of these multi-scale ice-ocean interactions would make CRISTAL unique in its capability to address climate issues of regional and relevance. Over oceans, which represents a secondary objective for the mission, the satellite will be able to measure sea surface height with an uncertainty of less than 3 cm. The main advantages and drawbacks of the Ka-band over the oceanic surface have been reviewed in Bonnefond et al (2018). Given its high along-track resolution of less than 10 km and high temporal resolution of sea level anomalies, the mission can further contribute a suite of sea level products including sea surface height and mean sea surface (vertical accuracy in sea level anomaly retrieval of less than 2 cm is requested). The radiometer on-board of CRISTAL corrects the satellite altimeter data for the excess path delay resulting from tropospheric humidity. The microwave radiometer measurements will complement wet tropospheric corrections derived from numerical weather prediction and non-collocated atmospheric data from other satellite instruments, to help meet the range accuracy requirement (Picard et al., 2015, Legeais et al., 2014, and Vieira et al., 2019).

Observation of water level at the (Arctic) coast as well as of rivers and lakes is a key quantity in hydrological research, (e.g. Jiang et al., 2017). Rivers and lakes not only supply freshwater for human use including agriculture but also maintain natural processes and ecosystems. The monitoring of global river discharge and its long-term trend contributes to the monitoring of global freshwater flux, which is critical for understanding the mechanism of global climate change. Satellite radar altimetry is a promising technology to do this on a regional to global scale. Satellite radar altimetry data has been used successfully to observe water levels in lakes and (large) rivers, and has also been combined with hydrologic/hydrodynamic models. Combined with gravity-based missions like NASA/DLR's GRACE and GRACE-FO, the joint use of the data will give information for ground water monitoring in the future.

## 4.5 Icebergs

Iceberg detection, volume change, and drift have been listed as a priority user requirement (Duchossois et al., 2018a; 2018b). Icebergs present a significant hazard to marine operations. Detection of icebergs in open water and in sea ice generally places a priority on wider satellite swaths to obtain greater geographic coverage. There is a need for automatic detection of icebergs for the safety of the navigation and chart production. Iceberg concentration is given in CMEMS' catalogue at 10 km resolution covering Greenland waters. SAR imagery is the core input for icebergs detection. However, iceberg detection (in particular small icebergs) is also possible using high-resolution altimeter waveforms. Tournadre et al. (2018) demonstrated detection of icebergs from CryoSat-2 altimeter data using several modes, and mention promising results with the Sentinel-3 data, which would result into a comprehensive dataset, already built under ALTIBERG project (Tournadre et al., 2016). The volume of an iceberg is valuable information for operational services and climate monitoring. For climate studies, the freshwater flux from

the volume of ice transported by icebergs is a key parameter, with large uncertainties related to the volume of the icebergs. Measuring volume is currently possible only with altimetry, by providing the iceberg freeboard elevation from the ocean surface. Iceberg volume has been calculated with altimetry with Envisat, Jason-1 and Jason-2 (e.g. Tournadre et al., 2015).

CryoSat-2 tracking over icebergs is operational but icebergs with high freeboards may be missed in the current range window. The range window definition for CRISTAL is defined in order to ensure that echoes from icebergs are correctly acquired. In-flight performances for the measurement of the Angle of Arrival from CryoSat-2 are around 25 arcsec. An equivalent performance is necessary to retrieve across-track slopes and elevations. The CRISTAL design of the instrument and the calibration strategy will be designed to comply with the specification of 20 arcsec. CRISTAL will provide the unprecedented

capability to detect icebergs at a horizontal resolution (gridded product) of at least 25 m. The products will be produced every 24 hours in synergy with other high-resolution data such as SAR imagery. Iceberg distribution and volume products will be produced at 50 km resolution (gridded) on a monthly basis.

## 4.6 Snow on land and permafrost

CRISTAL may support and contribute to studies and services in relation to seasonal snow cover and permafrost applications

over land. These are considered a secondary objective for the mission. The ability for the retrieval of snow depth with Ku-/Ka-band altimeter is limited over land (Rott et al. 2018). Snow studies over land area are so far largely limited to scatterometer in case of Ku-band, examples of such retrievals are reviewed in Bartsch (2010). Measurements as provided by CRISTAL may, however, be useful in retrieving internal properties of the snowpack such as existence of ice layers (e.g. due to rain on snow, Bartsch et al. 2010). The relevant properties of an upper snow layer contrast with that of an underlying ice layer (see also

section 4.2). Further, snow structure is reflected in differences observed in radar observations using different frequencies (Lemmetyinen et al. 2016). Snow structure anomalies as well as land surface state (freeze/thaw) are expected to be identified by time series analyses as such processes alter penetration depth. Altimeter data is also rarely used for permafrost studies. Such data can also be applied for monitoring lake level as proxy for permafrost change (Zakharova et al. 2017). Surface status is closely interlinked with ground temperature (e.g. Kroisleitner et al. 2018) but usage of satellite altimetry in this context remains

unexplored. Signal interaction with vegetation limits the applicability of Ku- and Ka-band for soil observations regarding freeze/thaw status (Park et. al. 2011) and also surface height. Wider use of altimetry for snow and permafrost applications requires higher spatial resolution and temporal coverage than available to date. An improvement regarding the latter issues is expected with CRISTAL, which will expand the utility of altimeter observations for permafrost and snow monitoring over land.

## 5 CRISTAL mission concept

This section summaries the envisaged primary payload components to address the CRISTAL mission objectives. The design draws from the experience of several in-orbit missions in addition to the ongoing developments within the Sentinel-6 and MetOp-SG programmes, and has a 7.5 years lifetime. CRISTAL's primary payload complement consist of:

- A **synthetic aperture radar (SAR) altimeter operating at Ku-band and Ka-band** centre frequencies for global elevation and topographic retrievals over land and marine ice, ocean and terrestrial surfaces (see Figure 2 and Figure 3). In Ku-band (13.5 GHz), the SAR altimeter can also be operated in interferometric (SARIn) mode to determine across-track echo location. The Ka-band channel (35.75 GHz) has been introduced to improve snow depth retrievals over sea ice, see e.g. Guerreiro et al (2016). A range (vertical) resolution of about 31 cm will be achieved to enhance freeboard measurement accuracy. Also, a high along-track resolution of about 20 m is envisaged to improve ice floe discrimination. Heritage missions include CryoSat-2 (SAR/Interferometric Radar Altimeter (SIRAL)), Sentinel-6 (Poseidon-4) and SARAL (AltiKa). The CRISTAL Altimeter (IRIS) is based on Poseidon-4 (Sentinel-6) and SIRAL (CryoSat-2) together with the addition of a Ka-band channel (analogous to AltiKa) and a bandwidth of 500MHz (at both frequencies) to meet the improved range resolution requirement in comparison to heritage altimeters. It has the capability for fully focused SAR processing for enhanced along track resolution by means of resolving full scatterer phase history (Egido and Smith, 2017). Digital processing will be implemented including matched filter range compression and on-board Range Cell Migration (RCM) compensation by means of a RMC mode for on-board data reduction (heritage from Poseidon-4) reducing downlink load. With respect to the dual frequency antenna (Ku- and Ka-band), an enhanced antenna mounting baseplate for improved baseline stability over CryoSat-2 will be required (20 arcsec vs ~30 arcsec for CryoSat-2).

- A high-resolution **passive microwave radiometer** is included with the capability to provide data allowing retrievals of total column water vapour over the global ocean and up to 10 km from the coast (by means of improving the measurement system with high frequency channels). The radiometer may also support cryosphere applications such as sea ice type classifications (Tran et al., 2009). Concerning the Microwave Instrument selection, potential options include: a potential US Custom Furnished Item based on the National Aeronautics and Space Administration (NASA)- Jet Propulsion Laboratory (JPL) AMR-C (Advanced Microwave Radiometer – Climate quality); development of an EU High Resolution radiometer solution; or a two-channel solution derived from the Sentinel-3 microwave radiometer. The feasibility of each of these options will be further evaluated in the next mission phase (Phase B2 at the time of the system Preliminary Design Review, expected late 2021).

- A **Global Navigation Satellite System (GNSS) receiver** compatible with both Galileo and Global Positioning System (GPS) constellations providing on-board timing, navigation and provision of data for on-ground precise orbit determination. Heritage GNSS solutions exist such as those based upon the GPS and Galileo compatible Sentinel-1,-

2,-3 C/D, Sentinel-6 A/B receivers. Precise Orbit Determination products will be provided by the Copernicus Precise Orbit Determination service.

• A **Laser Retro-reflector Array** (LRA) for use by the Satellite Laser Ranging network and by the International Laser Ranging Service for short arc validation of the orbit. Heritage concepts suitable for CRISTAL include CryoSat-2/Sentinel-3 LRAs.

Three modes of radar operation are envisaged, which are automatically selected depending on the geographic location over
the Earth's surface (see Table 3 and Figure 3), prioritising the retrieval of relevant geophysical parameters of interest:

• **Sea ice and iceberg mode**: In Figure 3, the proposed coverage is shown in orange. It is proposed that this mode makes a step forward in ice-thickness retrieval by operating the instrument with the SAR interferometer configuration in Ku-band, i.e. a two-antenna cross-track interferometric principle. The measurement mode will be in an open burst, or interleaved arrangement, in which receptions occur after each transmitted pulse. This results in an along-track
resolution by ground processing to up to a few metres, which enables small sea ice sheets to be distinguished and the detection of narrow leads between them. The disadvantages of the open burst transmission versus a closed burst operation mode include a larger data volume and the power demand and variations of the Pulse Repetition Frequency around the orbit. The interferometric operation allows the location of across-track sea ice leads, whilst open-burst timing allows full scatterer phase history re-construction for fully focussed processing (Egido and Smith, 2017). This
improves sea ice lead discrimination (by means of improvement in sampling and resolution), and hence retrievals of elevation, and thus polar Sea Level Anomalies (SLA) by a significant factor, Armitage and Davidson (2014). Open-burst Ka-band SAR is also provided to also allow for improving retrieval of snow depth over sea ice.

• **Land ice mode**: In Figure 3, the proposed coverage is shown in magenta. Land ice elevation is retrieved by means of improved surface tracking based on the large range window. The accuracy of elevation retrievals are likely improved
by a factor 2 by means of increasing the number of echoes per unit time by a factor 4 over the CryoSat-2 heritage design. The Ku-band SAR interferometer is used to retrieve the across-track point of closest approach supplemented with Ka-band SAR. Closed-burst operation (see e.g. Raney 1998) is used over this surface type, in which the reflections arriving back at the radar are received after each transmitted burst has finished.

• **Open and coastal ocean mode**: In Figure 3, the proposed coverage shown in magenta provides Arctic and southern
polar ocean retrieval of SLA and precision SAR altimetry to complement other ocean topography missions including Sentinel-3, Sentinel-6 and next generation topographic missions. In the case of open ocean, closed-burst SAR operation at Ku-band and Ka-band is used and the RMC on-board processing is applied, first implemented in the frame of Sentinel-6, which provides a considerable gain in instrument data rate reduction. In addition, data will be collected over inland water regions using one of the above modes.

**Table 3 Key altimeter characteristics in the different modes of operation (Credits: Thales Alenia Space, France).**

| | Open and Coastal Ocean | Sea Ice and Icebergs | | Land Ice | | |
| --- | --- | --- | --- | --- | --- | --- |
| | | Sea ice | Icebergs | Ice sheet interior (Ice sheet/ Ice caps) | Ice margin | Glaciers |
| $\sigma_0$ range in Ku-band | 6 dB to 25 dB | 0 dB to 55 dB | | 0 dB to +40 dB | -10 dB to +40 dB | -10 dB to +40 dB |
| $\sigma_0$ range in Ka-band | +8 dB to +27 dB | +2 dB to +57 dB | | 2 dB to +42 dB | -8 dB to +42 dB | -8 dB to +42 dB |
| Measurement mode in Ku-band | SAR Closed Burst | SARIn Interleaved | | SARIn Closed burst | | |
| Measurement mode in Ka-band | SAR Closed Burst | SAR Interleaved | | SAR Closed burst | | |
| Range window size | 256 points | 256 points | 256 points | 1024 points | 1024 points | 1024 points |
| Tracking window size | 256 points | 256 points | 256 points | 2048 points | N/A | N/A |
| Range window size | 64 m | 64 m | 64 m | 256 m | 256 m | 256 m |
| Tracking window size | 64 m | 64 m | 64 m | 512 m | N/A | N/A |
| Tracking mode | Closed loop | Closed loop | Closed loop | Closed loop | Open loop | Open loop |
| On-board processing | RMC | RMC | N/A | N/A | N/A | N/A |
| Optional On-board processing | Yes | N/A | N/A | N/A | N/A | N/A |

The latency of CRISTAL data products follows the requirements expressed in the PEG-1 and PEG-2 reports, and provides
measurements of different latencies according to the application need. The product latencies range from 3 hours (some ocean
L2 products), to 6 hours (sea ice freeboard products), 24 hours (sea ice thickness, sea ice snow depth, and iceberg detection
products), 48 hours (some ocean L1 and L2 products), and up to 30 days (surface elevation and some ocean L1 products).
These data latencies indicate the time interval from data acquisition by the instrument to delivery as a Level 1B data product
to the user.


# 6 Conclusions and CRISTAL mission status

CRISTAL directly addresses the EU Arctic Policy and primary user requirements collected by the European Commission and provides sustained, long-term monitoring of sea ice thickness and land ice elevations. It thereby responds to needs for continuous pan-Arctic altimetric monitoring including the region of the Arctic Ocean north of 81.5°N. Antarctica will be
equally well covered. The mission serves several key Copernicus operational services in particular the Climate Change service, Marine Environmental Monitoring Service and makes contributions to the Land Monitoring Service, Atmospheric Monitoring Service and Emergency Management Service.

CRISTAL will cover the polar regions with a Ku-band Interferometric Synthetic Aperture Radar Altimeter with supporting
Ka-band channel. In addition, the payload contains a high and low frequency passive microwave radiometer to perform wet troposphere delay correction, and surface-type classification over sea ice and ice sheets. The mission is designed for a 7.5 years design lifetime and will fly in an optimized orbit covering polar regions (omission <= 2°; weekly and monthly sub-cycles). A key element is the high along-track resolution (by ground processing up to a few meters when the novel interleaved SAR operation mode is used) to distinguish open ocean from sea ice surfaces. Thanks to the dual-frequency SAR altimetry
capability, a snow depth product will be produced over sea ice with high accuracy in response to long-standing user needs.

CRISTAL has undergone and completed parallel preparatory (Phase A/B1) system studies in which mission and system requirements have been investigated and consolidated. The intermediate system requirements review has been completed with parallel industrial consortia compliant with the mission and system requirements. Next steps include the full definition,
implementation and in-orbit commissioning of CRISTAL (Phases B2, C/D and E1) where a prototype and recurrent satellite will be developed.

*Author contributions.* M. Kern, as ESA Mission Scientist is responsible for the mission requirements for the CRISTAL mission and was responsible for the overall conceptualisation and structure of the paper. He drafted the manuscript and completed
revisions based on co-author contributions and review. R. Cullen led the CRISTAL technical activities and contributed to the system concept description in section 5. B. Berruti, J. Bouffard, T. Casal, M.R. Drinkwater, A. Gabriele, A. Lecuyot, M. Ludwig, R. Midhassel, I. Navas Traver, T. Parrinello, G. Ressler were involved in the supporting scientific and campaign activities or in the technical activities with industry. They contributed to sections 5 and 6 of this manuscript and to the overall pre-publication critical review of the work. E. Andersson and C. Martin-Puig described and provided input and critical review
of the Sections 1 and 2, which pertain mostly to the European Commission and EUMETSAT's involvement in the mission preparation and setup. O. Andersen, A. Bartsch, S. Farrell, S. Fleury, S. Gascoin, A. Guillot, A. Humbert, E. Rinne, A. Shepherd, M. R. van den Broeke, and J. Yackel were members of ESA's Mission Advisory Group in Phase A/B1 and provided input and critical review and assistance with the manuscript.

*Competing interests.* The authors declare that they have no conflict of interest.

*Acknowledgements.* The authors would like to acknowledge the industrial and scientific teams involved in the Phase A/B1 study of the CRISTAL mission significantly contributing to the success of the mission preparation in this feasibility phase. The authors would like to acknowledge the comments from three anonymous reviewers and the editor.


**Appendix A: List of acronyms**

| | | |
|---|---|---|
| | AltiKa | Altimeter Ka-band |
| | AMR-C | Advanced Microwave Radiometer-Climate Quality |
| | C3S | Copernicus Climate Change Service |
| 490 | Cal/Val | Calibration and Validation |
| | CAMS | Copernicus Atmospheric Monitoring Service |
| | CEMS | Copernicus Emergency Management Service |
| | CGLS | Copernicus Global Land Service |
| | CIMR | Copernicus Polar Passive Microwave Imaging Mission |
| 495 | CLS | Collecte Localisation Satellites |
| | CMEMS | Copernicus Marine Environmental Monitoring Service |
| | COP21 | United Nations Framework Convention on Climate Change, 21st Conference of the Parties |
| | CRISTAL | Copernicus Polar Ice and Snow Topography Altimeter |
| | CSC | Copernicus Space Component |
| 500 | dB | Decibel |
| | EC | European Commission |
| | EO | Earth Observation |
| | ESA | European Space Agency |
| | EU | European Union |
| 505 | EUMETSAT | EUropean Organisation for the Exploitation of METeorological SATellites |
| | FMI | Finnish Meteorological Institute |
| | GCOS | Global Climate Observing System |
| | GMES | Global Monitoring for Environment and Security |
| | GNSS | Global Navigation Satellite System |
| 510 | GPS | Global Positioning System |
| | IPCC | Intergovernmental Panel on Climate Change |
| | IRIS | Interferometric Radar altimeter for Ice and Snow |
| | JPL | Jet Propulsion Laboratory |
| | LRA | Laser Retro-reflector Array |
| 515 | MetOp-SG | Meteorological Operational Satellite - Second Generation |
| | NASA | National Aeronautics and Space Administration |
| | OCO | Open and coastal ocean |
| | OSTST2019 | Ocean Surface Topography Science Team Meeting 2019 |
| | PEG | Polar Expert Group |

| 520 | RADAR | Radio Detection and Ranging |
| | RCM | Rang Cell Migration |
| | RMC | Range Migration Compensation |
| | SAR | Synthetic Aperture Radar |
| | SARIn | Interferometric SAR |
| 525 | SARAL | Satellite with ARgos and ALtiKa |
| | SIRAL | SAR/Interferometric Radar Altimeter |
| | SLA | Sea Level Anomaly |
| | STC | Short Time Critical |

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

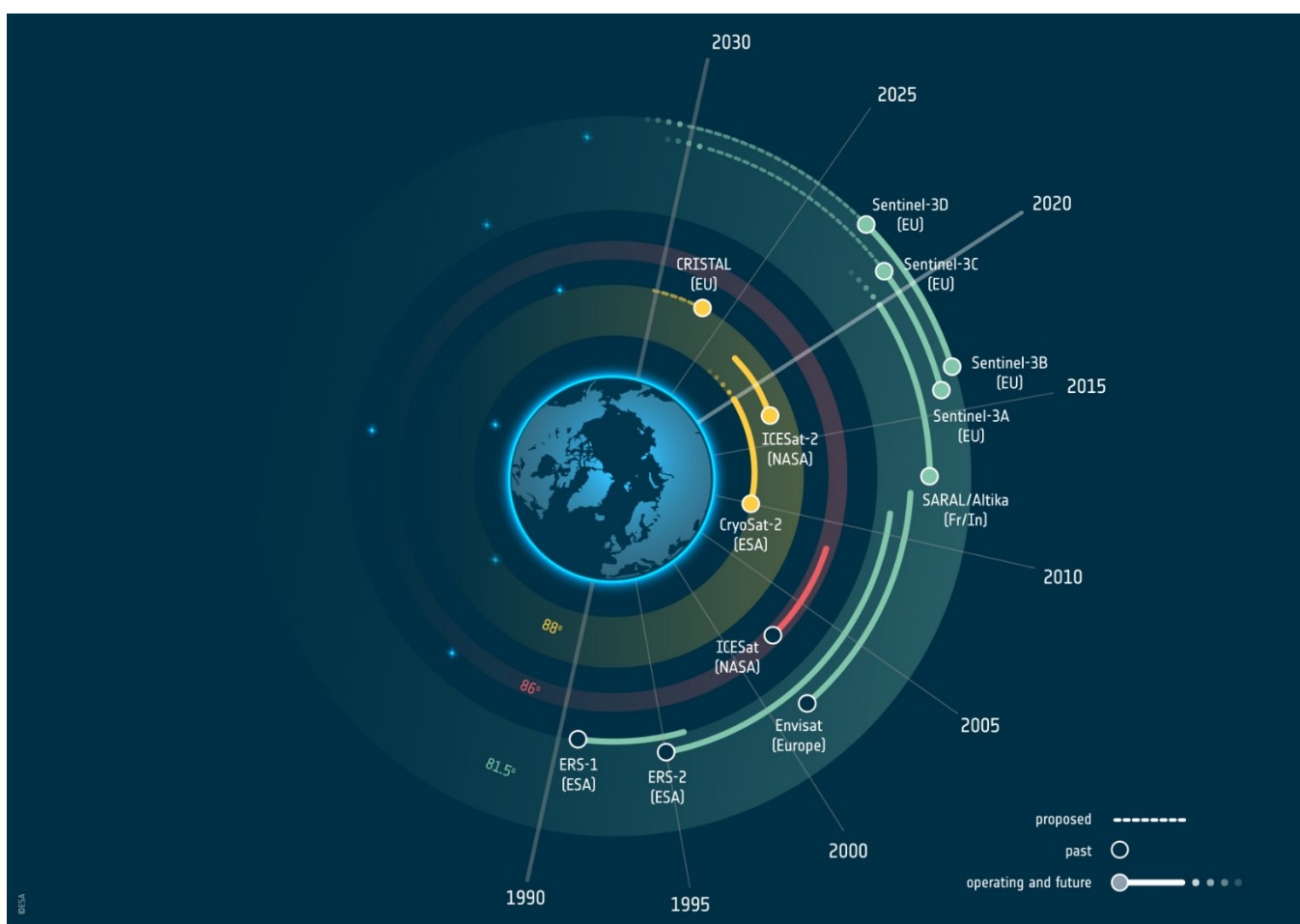

**Figure 1. Past, operating, approved and proposed polar topography altimeter missions. By mid 2020s, CRISTAL will fill the gap acquiring climate-critical data over polar ice north or south of 81.5° latitude (Image: EOGB/ESA).**

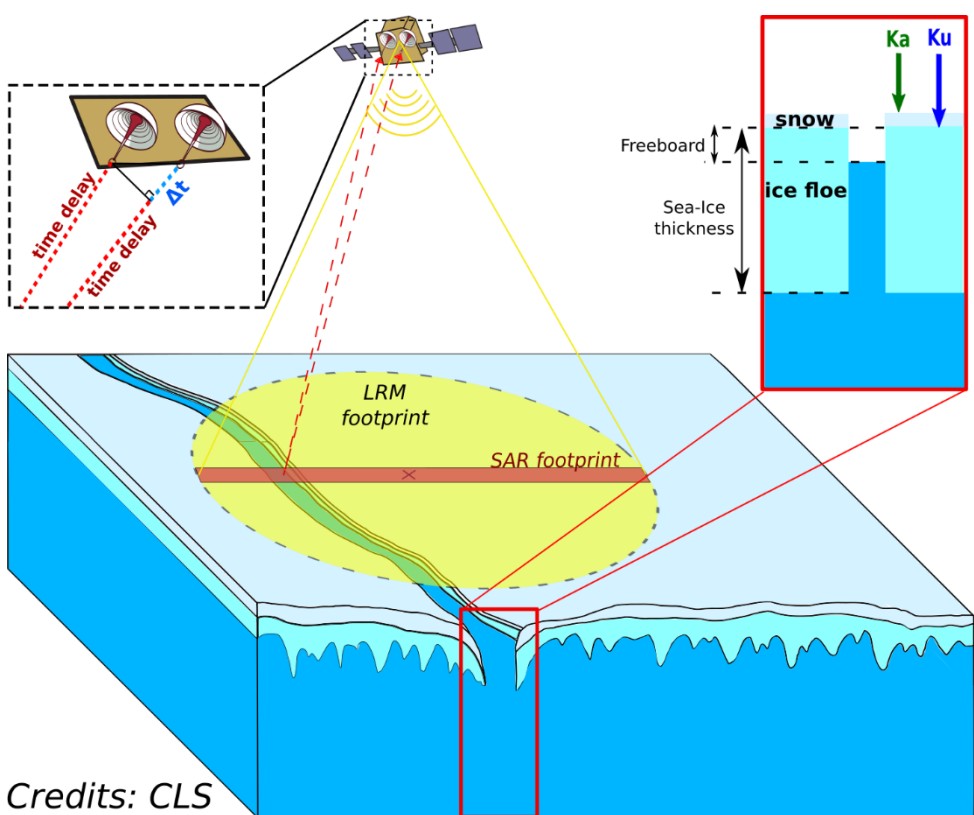

**Figure 2. Illustration of the CRISTAL observation concept over sea ice employing a twin-frequency, twin antenna SAR radar altimeter with interferometric capability at Ku-band (Image Credits: CLS).**

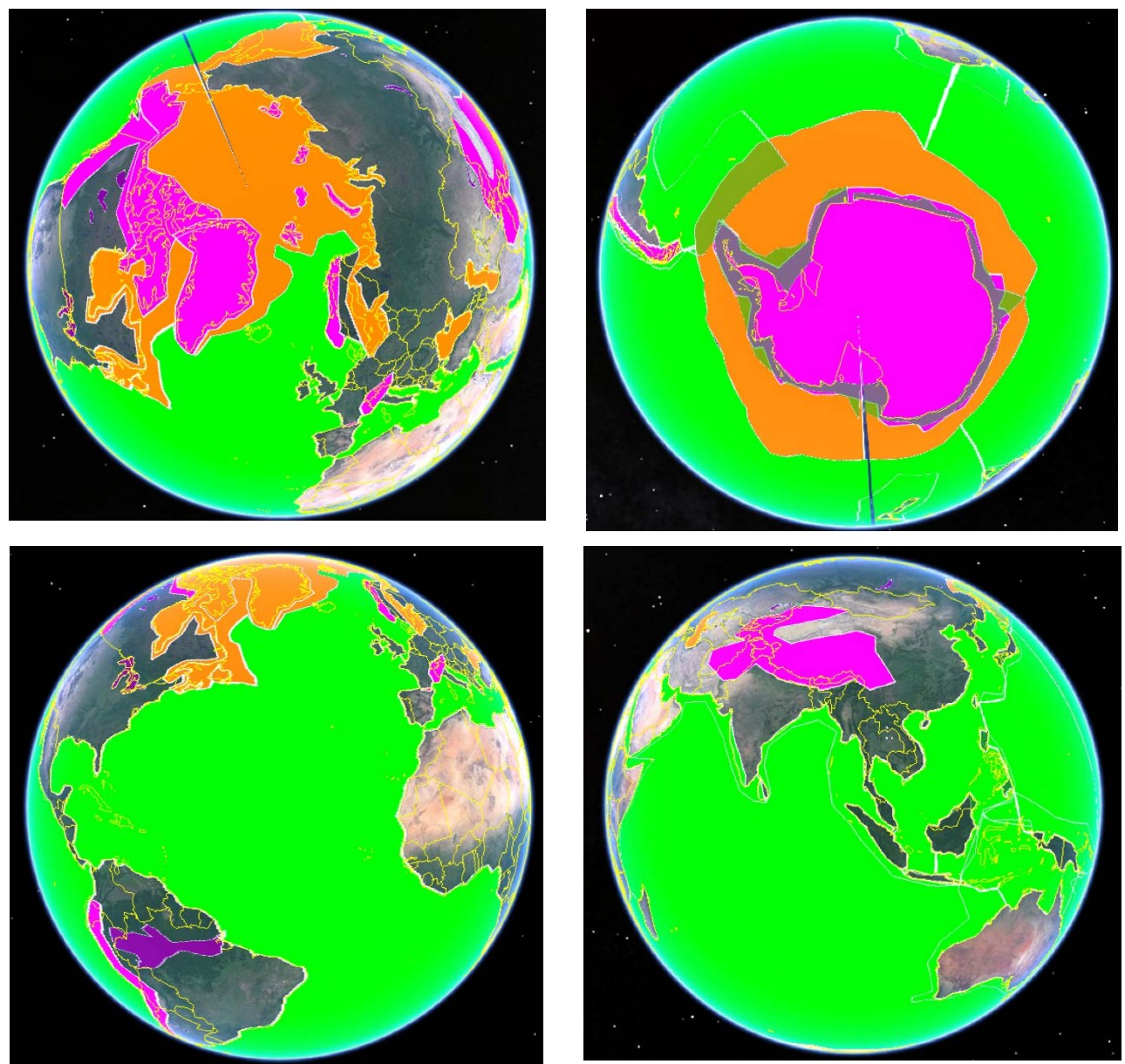

**Figure 3. Indicative mission geographic operating mode mask used in CRISTAL altimeter data volume sizing: Magenta = Land Ice closed-burst SARIn mode, also including smaller ice-caps; Orange = Sea-Ice and Icebergs open-burst SARIn mode (maximum coverage in North/Southern hemisphere); Green = Open and Coastal Ocean SARIn reduced window mode;**
**Purple = Inland water – this is not anticipated as a mode but may be derived from one of the three key modes. Note: The wedge type feature in some of the images is an artefact of the display software.**