# Peer review of "The Copernicus Polar Ice and Snow Topography Altimeter (CRISTAL) High-Priority Candidate Mission"

_The Cryosphere, 2020_

## Referee Comment (RC1) · Anonymous Referee #1 · 10 Mar 2020

——————— Summary ———————

This paper introduces the Copernicus polaR Ice and Snow Topography ALtimeter (CRISTAL), which is a Polar orbiting satellite that has been identified as a high-priority candidate mission by the European Commission (EC) in collaboration with the European Space Agency (ESA). CRISTAL will build on the heritage of previous Ku- and Ka-band satellite radar altimeters by operating at both frequencies, as well as flying a high-resolution passive microwave radiometer. Such a mission is crucial for the continuation and improvement of large-scale observations of the polar and terrestrial ice and snow cover, as well as ocean dynamics. Therefore, this paper will be of interest

to a large and varied readership, and I am pleased to see an update on CRISTAL's progress being submitted. However, I do have some concerns and comments that should be addressed before publication.

My main concern is that this should be "the" paper highlighting the importance of CRISTAL, above and beyond other candidate satellite missions. Therefore, it needs to be clear, convincing, and accessible to a wide audience. The current manuscript reads too much like a copy and paste from an ESA report.

- The paper would benefit from being more concise, with improved coherence between sections, less repetition, and an early focus on the scientific benefits of CRISTAL instead of having them spread throughout

- A number of acronyms are not introduced in the main text (e.g. GMES, EUMETSAT, OSTST), which assumes too much prior knowledge considering the wide readership potential

- It is very hard to digest such long sentences. I appreciate this is a style preference but it was an issue for me. This includes P2L43-46, P4 L120-122, P6170-186 (secondary objectives summary), P89L238-242, and many others.

——————— Specific comments ———————

P1L36: This makes it sound like the paper might be more technical than it is. Make it clear that the paper is primarily mission contributions, and does not include in-depth technical information (which can't be available at this time).

P3L89: What is meant by "next generation of the current Sentinels 1 to 6 series"? Could do with a little more information, or relevant references.

P4L105-106: What is "an integrated end-to-end system approach"? These more technical/agency terms should be explained in a science journal.

P4L13: Remove "inhospitable". The Arctic human population is mentioned in the same

sentence.

P5L150: Who recorded this recommendation? Please provide a reference.

Section 3: The beginning of Section 3 (up to P5L155) is very sea ice heavy. I encourage the authors to provide more on the importance (climatic and observationally) of glaciers, ice caps and ice sheets prior to introducing them as a primary mission objective

P6L179-180: Please provide some references for the evidence of frozen rivers and lakes being influenced by climate change

P7L202: The authors state that "Compared to heritage missions, the Ka-band channel (35.75 GHz) is added for snow..." but later in the paragraph, they describe SARAL (AltiKa) as a heritage mission, which could be confusing to readers who are not familiar with the history of Ka-band altimeters.

P7L208: Which radar system does the 500 MHz bandwidth apply to? As I read it, they mean just Ka. However, AltiKa also has a bandwidth of 500 MHz so I'm not sure how this would lead to an improved range resolution in comparison.

P7L209: The reference to Egido and Smith (2017) should also be included here

P8L239: Add reference to Armitage and Davidson (2013) – DOI: 10.1109/TGRS.2013.2242082

P8L244-245: The authors state that "Retrievals are likely improved by a factor 2..." but it's not clear what retrieval parameter they are referring to. The number of retrievals? Accuracy of individual retrievals?

P10L270-271: I understand that it is only Arctic sea ice that is a driving force of the global thermohaline circulation

P10L278: The Perovich (2017) reference is over two years out of date. NSIDC, for example, can provide the most up-to-date statistics on sea ice extent decline.

P11L311-313: Include some discussion/reference to Mallett et al. (2020) – DOI: 10.5194/tc-14-251-2020, which finds that assumptions concerning the time evolution of overlying snow density can lead to underestimates of sea ice thickness from radar altimetry. This will have the opposite impact of the salinity consideration of Nandan et al. (2017).

Section 5.1: Include a comment on the importance of sea ice in Antarctica. There are many examples relating to ecosystems/surface momentum exchange/ice shelf-ocean interactions etc.

Section 5.2: Currently this paragraph applies only to Arctic sea ice. The authors could address the difficulties of applying a dual-frequency snow depth retrieval method in Antarctica (much more complex penetration). Also, the first sentence needs tidying up.

P13L372: Add reference to Foresta et al. (2016) – DOI: 10.1002/2016GL071485

Sections 5.1 and 5.3 are lacking in references. This needs to be addressed before publication in a scientific journal.

P14L407: The designed operational lifetime of CRISTAL (7.5 years) is key and interesting information, so I suggest mentioning this earlier in the manuscript, such as in the introduction and even the abstract

P16L478-479: What is the timeframe of prototype and potential satellite development?

A couple of tables would be useful in the paper: One that summarizes the current mission milestones and timeframe, and another with instrument information (not limited to altimeters)

———— Technical comments ————

P2L41: "...see Chen et al (2013)" -> "...(Chen et al., 2013)"

P5L138: "...from SAR..." -> "...from SAR **altimetry**..."

P5L149: Remove "at large"

P6L180: "...context of global warming..." -> ""...context of climate change..."

P6L189: "requisite" -> "required"

P8L216: This opening bracket has no end

P9L250: Define SLA here, not P14L388

P9L252: RMC is already defined on P7L211

P9L261: "...delivery as **a** Level 1B..."

P11L286: "ice-infested" -> "ice-covered"

P13L370: "..horizontal resolution of less or equal than 100 m..." -> "horizontal resolution of less than or equal to 100 m..."

P13L378: "...helping us understanding and monitoring..." -> "...helping us to understand and monitor..."

P14L391: "...supporting **sea** ice thickness retrieval..."

P14L393: "associated to" -> "associated with"

General: Please be consistent between "sea-ice" and "sea ice" and the same for land ice

---

## Short Comment (SC1) · 12 Mar 2020

The CryoSat SARIn mode has been successful in demonstrating that coherent interferometric radar altimetry is better than traditional radar altimetry in monitoring change in glacial height, particularly at the periphery of the large ice caps and for glaciers and smaller ice caps. The problem remains that often the point-of-closest-approach (POCA) in these areas is at a cross-track look angle greater than that corresponding to a differential phase of -$\pi$ to +$\pi$ radians. For the CryoSat baseline this look angle is $\pm 0.54°$. A reference digital elevation model (DEM) can be used to help resolve the $2\pi$ phase ambiguity but height blunder errors can still exist, particularly with cross-track

slopes of greater than 1°. However, if the Ka-band channel on CRISTAL could be made interferometric then there is a straightforward approach which would remove the need for a reference DEM and allow an improved and more reliable mapping solution for glacial ice.

The advantages of this approach to the mapping of glacial height and height change are significant:

1. There is no need for a reference DEM.

2. There will be two solutions, both more reliable than that possible from a single frequency system, therefore, a more precise and accurate result. The Ka-band result in particular should be more accurate then the Ku.

3. The possibility of serious mapping errors which exist with a single frequency SIRAL-like system will be reduced.

4. The approach can also be used to improve the reliability of swath mode results.

5. Having two SARIn frequencies will also improve the ability to calibrate both systems.

Further details and an initial evaluation are given in the supplement. I hope that the CRISTAL programme is still at a stage where ESA can study this option more fully.

Please also note the supplement to this comment:
https://www.the-cryosphere-discuss.net/tc-2020-3/tc-2020-3-SC1-supplement.pdf

**Supplement:**

Comment on 'The Copernicus Polar Ice and Snow Topography Altimeter (CRISTAL): Expected Mission Contributions'

Laurence Gray, University of Ottawa, March 2020

The CryoSat SARIn mode has been successful in demonstrating that coherent interferometric radar altimetry is better than traditional radar altimetry in monitoring change in glacial height, particularly at the periphery of the large ice caps and for glaciers and smaller ice caps. The problem remains that often the point-of-closest-approach (POCA) in these areas is at a cross-track look angle greater than that corresponding to a differential phase of -π to +π radians. For the CryoSat baseline this look angle is ±0.54°. A reference digital elevation model (DEM) can be used to help resolve the 2π phase ambiguity but height blunder errors can still exist, particularly with cross-track slopes of greater than 1°. However, if the Ka-band channel on CRISTAL could be made interferometric then there is a straightforward approach which would remove the need for a reference DEM and allow an improved and more reliable mapping solution.

The steps in the process would be:

1.  Create three look-angle solutions using the Ku-band SARIn phase χ, and that phase ±2π.
2.  Calculate the equivalent Ka-band phase for each of the three Ku-band look angle solutions.
3.  Using the three Ku-band look angle solutions calculate the number of 2π's to be added or subtracted to the (wrapped) Ka phase from the Ka interferometric altimeter.
4.  Compare the three possible look-angle solutions from the corrected Ka phase above with the three Ku look-angle solutions. With a normal level of phase noise, one of the comparisons will match much more closely than the other two.

To demonstrate the potential of this approach, I have used the GIMP DEM of Greenland and randomly picked a sub-satellite track (Fig. 1; CryoSat descending pass on April 8, 2012) over the western periphery of the Greenland Ice Cap. The POCA position is calculated for each along-track sampling position allowing an estimation of the look-angles. Independent Gaussian distributed phase noise was added to simulate the POCA differential phase for both frequencies, calculated on the basis of the look-angles. The POCA height range along the line in Fig. 1 is from around 500 m to 2000 m and, as is normal here, many of the POCAs subtend a look-angle greater than 0.54°. The look-angle range is from 0.15° to 1.54°. The Fig. 1 upper plot illustrates the good agreement between the derived solution for the Ku and Ka look-angles. The middle plot illustrates the look-angle errors and shows that, as expected, the Ka error is significantly less than the Ku error. The lower plot illustrates the phase clustering for the three solutions and shows that the correct solution clustered around 0 can be readily picked. The standard deviation of added noise for this plot was 0.08 radians, which I think is larger than the phase noise in the processing scheme I use.

Further work could evaluate the extent to which random and bias phase errors, uncertainty in baseline attitude, and differences in radar response at the two frequencies could limit this approach. Note this approach is not new, two frequencies were used in the first demonstration of airborne repeat-pass interferometry and the movement of a radar reflector was recovered without concern of the multiple solutions due to 2π phase ambiguities (Gray and Marris-Manning, IEEE TGRS, 31, no. 1, pp 180-191, 1993).

[Figure]

Fig. 1 Left: Position of the track used to calculate the POCA look-angles superimposed on the GIMP DEM. Upper plot: Ku and Ka solutions for the POCA look-angles with (middle plot) the errors (Ka: red and Ku: blue). The lower plot illustrates the phase difference between the Ka-band phase derived from the three possible look-angle solutions and the Ka-band 'wrapped' phase.

The advantages of this approach to the mapping of glacial height and height change are very significant:

1. There is no need for a reference DEM.
2. There will be two solutions, both more reliable than that possible from a single frequency system, therefore, a more precise and accurate result. The Ka result in particular should be more accurate then the Ku.
3. The possibility of serious mapping errors which exist with a single frequency SIRAL-like system will be reduced.
4. The approach can also be used to improve the reliability of swath mode results.
5. Having two SARIn frequencies will also improve the ability to calibrate both systems.

I don't see any insurmountable complications re the satellite hardware. I assume that there will be a spare Ka receiver. I also assume there will be a composite multi-frequency horn feed, this will have to illuminate only part of the two parabolic dishes at Ka-band as there would be the requirement that the cross-track antenna pattern be comparable to that at Ku-band. This may require a re-think of the Ka-band link budget and the Ka burst frequency must adequately sample the azimuth Doppler bandwidth. I hope that the CRISTAL programme is still at a stage where ESA can study this option. If it can be implemented, it will make a huge improvement to the utility of CRISTAL for mapping of change in glacial ice. NASA has made a significant step forward in going from IceSAT-1 to IceSAT-2, ESA can make a comparable step forward in monitoring change in glacial ice around the world with the CRISTAL satellite if this option can be implemented.

---

## Referee Comment (RC2) · Anonymous Referee #2 · 24 Mar 2020

The paper presents an introduction and major update to the anticipated satellite mission, Copernicus Polar Ice and Snow Topography Altimeter (CRISTAL). The key observables of the mission are sea-ice thickness and elevation measurements of ice sheets and glaciers. The motivation for these observables are described. The content of the paper is useful for the scientific and broader communities, and warrants publication. However, there are major issues in the clarity, organization, and redundancy of the text that require attention. These issues prevent the main points of the paper from being communicated.

My major recommendations are to: consolidate Section 5 with the mission objectives

listed in Section 3 since much of the content is repetitive; break up numerous run-on sentences and rewrite for clarity; carefully proof the manuscript and correct the many typos and grammatical errors within.

L24. Here and throughout the paper, it is not clear what is meant by "evolution." Please describe.

L28. "properties of snow cover on ice" I recommend changing this to snow depth since its retrieval has been demonstrated with Ku- and Ka-band data, whereas the other suggested property retrievals have not yet been fully demonstrated.

- Polar Regions, Polar Oceans, and Total Column Water Vapour should not be capitalized.

L29. "foreseen" does not seem like the correct word here since it is not a prediction. "Planned" may be a better word.

L43-46. Here and throughout the paper, there are run-on sentences with mixed messages. Please rewrite these for clarity.

L50-51. I'd suggest changing this to stakeholders for inclusivity. Derived products may be useful for indigenous communities.

L51. While I understand this is a European effort, the listed motivations in this paragraph also apply to other countries. I'd suggest broadening this or adding a statement to be more inclusive of the international community, as it would strengthen the motivation and utility of such a mission.

L53. "It also has..." what does "it" refer to here?

L57. These potential impacts are not limited to European weather. They may affect global weather patterns.

L61-63. The following section is more about the history of the Copernicus programme, rather than an overview of the missions under development. I'd suggest editing either

this sentence or the section to be consistent with one another. Depending on who the target audience is, the mission overviews may be more useful for the broader community than the history of the Copernicus programme.

L70-82. Who is the target audience of this paper? Do they need to know the history of the Copernicus programme?

L84. This is vague. Potential for what?

L85. For future what? This is vague.

L103. What "so-called Long-Term Scenario" is this sentence referring to? It is unclear.

L105. "EC" and "RMC" elsewhere. Use acronyms consistently.

L105-106. What does "integrated end-to-end system approach" mean? Here and throughout, such buzzwords are not informative.

L133. UNFCC. This and other acronyms are defined and used only once. I recommend deleting unnecessary acronyms such as this one for easier reading.

L150. OSTST in-text definition missing.

L155+ I suggest using bold font or italics for the first lines of each bullet point for easier reading.

L202-203. "is added for snow depth measurements to distinguish between snow and ice layers" needs rewriting for clarity.

L216-217. Is a parenthesis missing?

L242. Add "over sea ice" after snow depth.

L254. What does "on a best effort basis" mean?

L272. "its presence limits human access" I suggest considering a different perspective of sea ice. It's a platform enabling subsistence hunting and travel for indigenous coastal

communities.

L274. ice-infested. Why use a word with such negative connotations?

L280-281. "to safeguard both climate and operational data services" Safeguard does not seem like the right word. "extending" or "advancing" may be better.

L282-289. Confusing, run-on sentences.

L288. Essential Climate Variable sounds important, but what does that really mean?

L308. Please clarify that this information relates to Copernicus sea ice thickness products.

L310. Please state which satellite this uncertainty pertains to.

L319-322. Run-on, confusing sentence.

L320. This uncertainty value is not consistent with the one given in the preceding paragraph. Please provide more detail on why these are different values.

L330-333. References are needed. The uncertainty in snow depth data from the historical period is as good as it can possibly get, on the order of 1 cm and less. The uncertainty is not halved over first year sea ice. Several studies have shown snow to be thinner over first year sea ice in areas where Operation IceBridge surveys were conducted. In other regions of the Arctic, deep snow can exist on first year sea ice.

L333 Giles et al. 2007 seems like a more appropriate reference here since it was the first study to demonstrate the propagating uncertainties associated with snow depth and other geophysical parameters.

L338-341. Please rewrite for grammar.

L354-355. Computing mass balance and identifying mass imbalance seem like the same thing here.

L360. The continuous record provides.... a long-term record. This is circular.

L365-366. Are commas missing? Please rewrite for grammar.

L375. "agility of tracking" It's not clear what is meant by this. How is the satellite going to be agile?

L378. Grammar.

L382-383. I suggest adding a little more detail here for clarity, e.g. retrieval accuracy of what exactly?

L386. "allow" seems like the wrong word here. "Have" may be better.

L389. It would be relevant to state the anticipated spatial resolutions here since leads come in all sizes.

L399-401. This is unclear. Please rewrite.

L403. Reference needed for main causes...

L404. Reference needed for largest uncertainty in the current prediction...

L414. Please rewrite for clarity.

L427-428. These sentences are vague. Please be more direct. Is snow depth retrieval not possible over land with CRISTAL frequencies?

L432-433. What is meant by status?

L434-435. Is it the spatial resolution that limits the wide use of altimetry data for snow and permafrost research or the frequencies used?

L460+ Wouldn't it be important to mention the relevance for monitoring the Antarctic cryosphere?

L467. Is this true? ICESat-2 reaches 88-deg latitude.

L471. It'd be helpful to restate the along-track resolution here since it is a key element for the mission.

---

## Referee Comment (RC3) · Anonymous Referee #3 · 26 Mar 2020

Overview: This paper presents the justification for and early design candidate for the CRISTAL mission. The paper also presents on the individual science applications that will be serviced by such a mission.

Summary: Firstly, apologies for my tardy review, I could would like to blame it on the new situation the world finds itself in with COVID-19 but I procrastinated well before that became a factor.

A thorough presentation of the need for, and technical capabilities of, the CRISTAL mission is a welcomed contribution to the community. The mission is very exciting and will provide invaluable data necessary to advance polar sciences in the coming

decades.

Overall, I found this manuscript very difficult to review. I tried to review the manuscript multiple times but the first 4 sections were such a slog that it was unmotivating. The first 4 sections read like a cut and paste of multiple science agency white papers while the final section, section 5, provides a nice science justification for the mission but is completely decoupled from the first 4 sections. In my opinion the paper requires significant restructuring and more consideration as to what is included and why.

I would start with a section (1) that provides the motivation for the mission: that is the science justification (essentially section 5) with a dedicated subsection that summarizes the EU Arctic Policy and primary user requirements that motivate the mission. (2) I would then summarize the geophysical observables that are needed to satisfy the science and application needs (maybe as a table). There is a table for radar specification but not for the geophysical observation requirements that are more relevant to an overview paper. This could be followed by a section (3) that describes the CRISTAL mission capabilities showing that it meets the needs presented in the prior section. The current section 2 could be reduced to 1 or 2 paragraphs that get included in the new section (3), no need to review the full Copernicus program. I think nearly all of the material is in the document, it just needs to be reworked into a coherent justification and description of the mission.

I have many small comments on grammar and wording but I don't think the manuscript is at the point yet where those details would be all that useful. For that reason, I only list my major comments here: - There is a reasonable amount of redundant text between section 5 and the rest of the manuscript. This is mostly a result of organization and can be fixed by structuring and being more selective on what's included - Sections 1-4 lack references supporting statements (I believe there is only 1 reference supporting the idea that the combination of ka- and ku- band altimeter will substantially improve measurements of snow on sea ice. Also several citations are outdated (e.g. Chen 2013), DeCanto and Pollard, 2016 is highly controversial and is not a representative citation for a current best estimate of future ice sheet change. - There is virtually no justification for including a microwave radiometer and no supporting literature, this should be shored up with a clear justification as to the applications (with list of geophysical observables) and science it will support with appropriate citations - Observing mode "Land-ice and Glacier" doesn't make a lot of sense since glaciers are Land-ice. I would suggest renaming this mode to "Land-ice" -or- glaciers and permafrost (GP) -or- glaciers, ice sheets and permafrost (GISP) - It is not clear how CRYSTAL will handle range ambiguities over complex glacier and permafrost terrain as advertised. - The colors for Figure 3 are difficult to distinguish (particularly magenta and purple). The figure should also include a colorbar and pole hole data gap.

---

## Author Comment (AC1) · 20 Apr 2020

Please see our detailed responses in blue color in the reviewers text in the attached supplement pdf.

Please also note the supplement to this comment:
https://www.the-cryosphere-discuss.net/tc-2020-3/tc-2020-3-AC1-supplement.pdf

———————————————

---

## Author Comment (AC2) · 20 Apr 2020

**16 March 2020**

Dear Laurence

Thank you for this careful review and your suggestion on the introduction of Ka-band SARIn for CRISTAL over land ice in addition to its current capabilities.

The capabilities and needs for the altimeter, in particular for what concerns the interferometric mode, have been extensively studied in Phase A/B1 by the industrial teams and discussed with the Mission Advisory Group. The Ka-band channel has mainly been introduced to improve snow depth retrievals over sea ice, for which interferometry is not required.

You are correct to note that the addition of a Ka-band interferometer would obviate the need for a reference elevation model when resolving phase ambiguities present in Ku-band interferometric data.

However, although phase ambiguities occur in areas of rugged terrain, it has been shown (e.g. Helm et al., 2014; McMillan et al., 2014; McMillan et al., 2016) that the use of a single frequency Ku-band interferometric altimeter allows for the determination of ice sheet elevation and elevation change to an accuracy that greatly exceeds the measurement requirements identified in the mission planning stages, and so in this regard a second interferometer is surplus to requirements.

Moreover, the addition of a second interferometer has been assessed to be a significant burden to the mission budget and schedule, presenting a risk to the availability of polar altimetry in the 2020's.

For these reasons, a second Ka-band interferometer is no longer part of the CRISTAL mission design.

Best regards,
Michael Kern, Robert Cullen, Bruno Berruti, Jerome Bouffard2, Tania Casal, Mark R. Drinkwater, Antonio Gabriele, Arnaud Lecuyot, Michael Ludwig, Rolv Midthassel, Ignacio Navas Traver, Tommaso Parrinello, Gerhard Ressler, Erik Andersson, Cristina Martin-Puig, Ole Andersen, Annett Bartsch, Sinead Farrell, Sara Fleury, Simon Gascoin, Amandine Guillot, Angelika Humbert, Eero Rinne, Andrew Shepherd, Michiel R. van den Broeke, John Yackel

---

## Author Response (AR1)

**Response to Anonymous Referee #1, received and published: 10 March 2020**

**13 March 2020**

Please see our responses in blue color in the reviewers text.

Thank you for this careful review and the constructive comments below, which have been used to revise the paper.

**Anonymous Referee #1**

──────── Summary ────────
This paper introduces the Copernicus polaR Ice and Snow Topography ALtimeter (CRISTAL), which is a Polar orbiting satellite that has been identified as a high-priority candidate mission by the European Commission (EC) in collaboration with the European Space Agency (ESA). CRISTAL will build on the heritage of previous Ku- and Ka-band satellite radar altimeters by operating at both frequencies, as well as flying a high-resolution passive microwave radiometer. Such a mission is crucial for the continuation and improvement of large-scale observations of the polar and terrestrial ice and snow cover, as well as ocean dynamics. Therefore, this paper will be of interest to a large and varied readership, and I am pleased to see an update on CRISTAL's progress being submitted. However, I do have some concerns and comments that should be addressed before publication.

Thank you for this careful review and the constructive comments.

My main concern is that this should be "the" paper highlighting the importance of CRISTAL, above and beyond other candidate satellite missions. Therefore, it needs to be clear, convincing, and accessible to a wide audience. The current manuscript reads too much like a copy and paste from an ESA report.

The paper is intended to inform the scientific and user community of this candidate mission in preparation. It is not the intention of the authors to rank or priortise this candidate mission above and beyond other candidate satellite missions in preparation. To address the comment from the reviewer, it is proposed to change the title, the abstract and to restructure/reorganise the paper has been considerably so that it reads better. We have tried to rework the paper so that it reads less like a white paper.

- The paper would benefit from being more concise, with improved coherence between sections, less repetition, and an early focus on the scientific benefits of CRISTAL instead of having them spread throughout

The mission is addressing operational user requirements and needs identified by the European Commission. A general background and context of the mission is considered important as it cannot be found anywhere else.

However, the authors acknowledge the concern of the reviewer and propose a considerable change in the paper (new title, now "The Copernicus Polar Ice and Snow Topography Altimeter (CRISTAL) High-Priority Candidate Mission", a better tailored abstract and to rework the paper (e.g. interchange sections 4 and 5).

- A number of acronyms are not introduced in the main text (e.g. GMES, EUMETSAT, OSTST), which assumes too much prior knowledge considering the wide readership Potential

Ok. We have checked again the acronym list and tried to introduce them all at the first instance in the text. Also, a large number of acronyms are removed for clarity.

- It is very hard to digest such long sentences. I appreciate this is a style preference but it was an issue for me. This includes P2L43-46, P4 L120-122, P6170-186 (secondary objectives summary), P89L238-242, and many others.

Ok. We have re-arranged all sentences mentioned by the reviewer. We have also reworked the paper so that it becomes better readable.

————— Specific comments —————

P1L36: This makes it sound like the paper might be more technical than it is. Make it clear that the paper is primarily mission contributions, and does not include in-depth technical information (which can't be available at this time).
Ok. Modified.

P3L89: What is meant by "next generation of the current Sentinels 1 to 6 series"? Could do with a little more information, or relevant references.
Ok, reference added.

P4L105-106: What is "an integrated end-to-end system approach"? These more technical/ agency terms should be explained in a science journal.
Ok. A sentence is added.

P4L13: Remove "inhospitable". The Arctic human population is mentioned in the same sentence.

OK. done

P5L150: Who recorded this recommendation? Please provide a reference.

The OSTST in their closing session. Reference added.

Section 3: The beginning of Section 3 (up to P5L155) is very sea ice heavy. I encourage the authors to provide more on the importance (climatic and observationally) of glaciers, ice caps and ice sheets prior to introducing them as a primary mission objective.
Agreed. Floating ice are listed as the top priority in the PEG report. Therefore, this is more emphasised here. The importance of glaciers and ice caps is now further emphasised.

P6L179-180: Please provide some references for the evidence of frozen rivers and lakes being influenced by climate change
ok, new reference added as well as sentence slightly modified.

P7L202: The authors state that "Compared to heritage missions, the Ka-band channel (35.75 GHz) is added for snow: : :" but later in the paragraph, they describe SARAL (AltiKa) as a heritage mission, which could be confusing to readers who are not familiar with the history of Ka-band altimeters.
Ok, sentenced changed.

P7L208: Which radar system does the 500 MHz bandwidth apply to? As I read it, they mean just Ka. However, AltiKa also has a bandwidth of 500 MHz so I'm not sure how this would lead to an improved range resolution in comparison.
500 MHz applies to entire system and both frequencies, Ka-band and Ku-band. This is now better formulated.

P7L209: The reference to Egido and Smith (2017) should also be included here
Ok. Added.

P8L239: Add reference to Armitage and Davidson (2013) – DOI: 10.1109/TGRS.2013.2242082
Ok, added, Armitage and Davidson (2014).

P8L244-245: The authors state that "Retrievals are likely improved by a factor 2: : :" but it's not clear what retrieval parameter they are referring to. The number of retrievals? Accuracy of individual retrievals?

Ok, updated.

P10L270-271: I understand that it is only Arctic sea ice that is a driving force of the global thermohaline circulation

ok, changed.

P10L278: The Perovich (2017) reference is over two years out of date. NSIDC, for example, can provide the most up-to-date statistics on sea ice extent decline.

Ok, added.

P11L311-313: Include some discussion/reference to Mallett et al. (2020) – DOI: 10.5194/tc-14-251-2020, which finds that assumptions concerning the time evolution of overlying snow density can lead to underestimates of sea ice thickness from radar altimetry. This will have the opposite impact of the salinity consideration of Nandan et al. (2017).

OK. Text added and Reference added. Please note that the Mallett et al reference was not available at the time of submission of this paper.

Section 5.1: Include a comment on the importance of sea ice in Antarctica. There are many examples relating to ecosystems/surface momentum exchange/ice shelf-ocean interactions etc.

Ok, one sentence is added.

Section 5.2: Currently this paragraph applies only to Arctic sea ice. The authors could address the difficulties of applying a dual-frequency snow depth retrieval method in Antarctica (much more complex penetration). Also, the first sentence needs tidying up.

First sentence corrected. Complex situation over Antarctica mentioned.

P13L372: Add reference to Foresta et al. (2016) – DOI: 10.1002/2016GL071485
Sections 5.1 and 5.3 are lacking in references. This needs to be addressed before publication in a scientific journal.

Foresta reference is added. It is noted that Section 5.1 now contains almost 25 references, Section 5.3 8 references. Please understand that the paper cannot provide a complete overview of all aspects studied in literature wrt sea ice and ice sheets; it is also not the intention of this paper but to introduce CRISTAL.

P14L407: The designed operational lifetime of CRISTAL (7.5 years) is key and interesting information, so I suggest mentioning this earlier in the manuscript, such as in the introduction and even the abstract

It is now added earlier in chapter 4.

P16L478-479: What is the timeframe of prototype and potential satellite development? A couple of tables would be useful in the paper: One that summarizes the current mission milestones and timeframe, and another with instrument information (not limited to altimeters)

This information cannot be provided at this point in time, as it is now yet known and/or depends on the consortiums and instrumentation selected for Phase B2, C/D, E1. More details on the instrument algorithms and performance could be subject for future publications.

———— Technical comments ————
P2L41: ": : :see Chen et al (2013)" -> ": : :(Chen et al., 2013)"

Ok, corrected.

P5L138: ": : :from SAR: : :" -> ": : :from SAR **altimetry**: : :"

Ok, corrected.

P5L149: Remove "at large"

Ok, corrected.

P6L180: ": : :context of global warming: : :" -> "": : :context of climate change: : :"

Ok, corrected.

P6L189: "requisite" -> "required"

Ok, corrected.

P8L216: This opening bracket has no end

Ok, corrected.

P9L250: Define SLA here, not P14L388

OK, introduced earlier.

P9L252: RMC is already defined on P7L211

OK, corrected.

P9L261: ": : :delivery as **a** Level 1B: : :"

OK, corrected.

P11L286: "ice-infested" -> "ice-covered"

OK, corrected.

P13L370: "..horizontal resolution of less or equal than 100 m: : :" -> "horizontal resolution of less than or equal to 100 m: : :"

OK

P13L378: ": : :helping us understanding and monitoring: : :" -> ": : :helping us to understand and monitor: : :"

Ok, corrected.

P14L391: ": : :supporting **sea** ice thickness retrieval: : :"

Ok, corrected.

P14L393: "associated to" -> "associated with"

Ok, corrected.

General: Please be consistent between "sea-ice" and "sea ice" and the same for land Ice

OK, corrected.

**Response to Anonymous Referee #2, received and published: 24 March 2020**

**27 March 2020**

Please see our responses in blue color in the reviewers text.

Thank you for this careful review and the constructive comments below, which have been used to revise the paper.

**Anonymous Referee #2**
The paper presents an introduction and major update to the anticipated satellite mission, Copernicus Polar Ice and Snow Topography Altimeter (CRISTAL). The key observables of the mission are sea-ice thickness and elevation measurements of ice sheets and glaciers. The motivation for these observables are described. The content of the paper is useful for the scientific and broader communities, and warrants publication. However, there are major issues in the clarity, organization, and redundancy of the text that require attention. These issues prevent the main points of the paper from being communicated.

Thank you for your suggestions and comments. We have re-organised the paper to take into account the reviewers comments by changing the title, the abstract, shortened section 2 considerable and interchanged sections 4 and 5. These structural changes help to reduce repetition.

My major recommendations are to: consolidate Section 5 with the mission objectives listed in Section 3 since much of the content is repetitive; break up numerous run-on sentences and rewrite for clarity; carefully proof the manuscript and correct the many typos and grammatical errors within.

Thank you. A re-organisation of the text has been done to account for this.

L24. Here and throughout the paper, it is not clear what is meant by "evolution." Please describe.
The term 'evolution' is now explained within abstract text, see line 30.

L28. "properties of snow cover on ice" I recommend changing this to snow depth since its retrieval has been demonstrated with Ku- and Ka-band data, whereas the other suggested property retrievals have not yet been fully demonstrated.
Ok, changed.

- Polar Regions, Polar Oceans, and Total Column Water Vapour should not be capitalized.
Ok, all capitalisations corrected.

L29. "foreseen" does not seem like the correct word here since it is not a prediction. "Planned" may be a better word.
Ok, changed to 'planned'.

L43-46. Here and throughout the paper, there are run-on sentences with mixed messages. Please rewrite these for clarity.
The sentence has been changed and shortened.

L50-51. I'd suggest changing this to stakeholders for inclusivity. Derived products may be useful for indigenous communities.
Ok, changed to stakeholders.

L51. While I understand this is a European effort, the listed motivations in this paragraph also apply to other countries. I'd suggest broadening this or adding a statement to be more inclusive of the international community, as it would strengthen the motivation and utility of such a mission.
Ok, understood and changed to be more inclusive.

L53. "It also has..." what does "it" refer to here?

Sentence removed. It referred to 'Europe'.

L57. These potential impacts are not limited to European weather. They may affect global weather patterns.
Ok, changed and 'European' removed.

L61-63. The following section is more about the history of the Copernicus programme, rather than an overview of the missions under development. I'd suggest editing either this sentence or the section to be consistent with one another. Depending on who the target audience is, the mission overviews may be more useful for the broader community than the history of the Copernicus programme.
Ok, the sentence was changed.

L70-82. Who is the target audience of this paper? Do they need to know the history of the Copernicus programme?
Ok, agreed. We have shortened the first two paragraphs in this section and merged them. We need some introductory sentences since otherwise we cannot introduce the concept of the HPCMs.

L84. This is vague. Potential for what?
Ok, shortened and changed to 'The intense use of Copernicus'.

L85. For future what? This is vague.
Ok, removed.

L103. What "so-called Long-Term Scenario" is this sentence referring to? It is unclear.
OK, it is now explained.

L105. "EC" and "RMC" elsewhere. Use acronyms consistently.
Ok, changed use of EC and RMC consistently.

L105-106. What does "integrated end-to-end system approach" mean? Here and throughout, such buzzwords are not informative.
The paragraph was removed.

L133. UNFCC. This and other acronyms are defined and used only once. I recommend deleting unnecessary acronyms such as this one for easier reading.
Ok, acronym UNFCC and several others removed from the list.

L150. OSTST in-text definition missing.
Ok, added in the text.

L155+ I suggest using bold font or italics for the first lines of each bullet point for easier reading.
OK, used 'bold font' to better highlight these lines.

L202-203. "is added for snow depth measurements to distinguish between snow and ice layers" needs rewriting for clarity.

Ok, sentence re-written.

L216-217. Is a parenthesis missing?
Ok, corrected.

L242. Add "over sea ice" after snow depth.
Ok, corrected.

L254. What does "on a best effort basis" mean?
It will be systematically observed if data volume /downlink capabilities allow so. "On a best effort basis" removed.

L272. "its presence limits human access" I suggest considering a different perspective of sea ice. It's a platform enabling subsistence hunting and travel for indigenous coastal communities.
Ok, sentence changed.

L274. ice-infested. Why use a word with such negative connotations?
Ok, removed.

L280-281. "to safeguard both climate and operational data services" Safeguard does not seem like the right word. "extending" or "advancing" may be better.
Ok, we changed it to 'extending'.

L282-289. Confusing, run-on sentences.
Ok, sentences shortened and changed.

L288. Essential Climate Variable sounds important, but what does that really mean?
It is a term used by GCOS. We will not change it in the text.

L308. Please clarify that this information relates to Copernicus sea ice thickness products.
Rewritten to 'Most sea ice thickness products …'

L310. Please state which satellite this uncertainty pertains to.
CryoSat-2 data. Added to the text.

L319-322. Run-on, confusing sentence.
Shortened and corrected.

L320. This uncertainty value is not consistent with the one given in the preceding paragraph. Please provide more detail on why these are different values.
Paragraph was updated and more details provided.

L330-333. References are needed. The uncertainty in snow depth data from the historical period is as good as it can possibly get, on the order of 1 cm and less. The uncertainty is not halved over first year sea ice. Several studies have shown snow to be thinner over first year sea ice in areas where Operation IceBridge surveys were conducted. In other regions of the Arctic, deep snow can exist on first year sea ice.
Another reference added and the text was shortened to make it more readable.

L333 Giles et al. 2007 seems like a more appropriate reference here since it was the first study to demonstrate the propagating uncertainties associated with snow depth and other geophysical parameters.
Ok, the reference was added and the sentence shortened.

L338-341. Please rewrite for grammar.
Ok, sentences updated and changed.

L354-355. Computing mass balance and identifying mass imbalance seem like the same thing here.
Sentence updated and changed.

L360. The continuous record provides.... a long-term record. This is circular.
Ok, sentence changed.

L365-366. Are commas missing? Please rewrite for grammar.
Commas added. Sentences shortened.

L375. "agility of tracking" It's not clear what is meant by this. How is the satellite going to be agile?
Sentence changed.
L378. Grammar.
Changed.
L382-383. I suggest adding a little more detail here for clarity, e.g. retrieval accuracy of what exactly?
Ok, changed.

L386. "allow" seems like the wrong word here. "Have" may be better.
Ok, changed.

L389. It would be relevant to state the anticipated spatial resolutions here since leads come in all sizes.
Ok, added.

L399-401. This is unclear. Please rewrite.
L403. Reference needed for main causes...
Reference added. Shepherd, A., Fricker, H.A. & Farrell, S.L. Trends and connections across the Antarctic cryosphere. Nature 558, 223–232 (2018). https://doi.org/10.1038/s41586-018-0171-6
L404. Reference needed for largest uncertainty in the current prediction.
Reference added. Edwards, T.L., Brandon, M.A., Durand, G. et al. Revisiting Antarctic ice loss due to marine ice-cliff instability. Nature 566, 58–64 (2019). https://doi.org/10.1038/s41586-019-0901-4.
L414. Please rewrite for clarity.
Ok, sentence changed.
L427-428. These sentences are vague. Please be more direct. Is snow depth retrieval not possible over land with CRISTAL frequencies?
Text modified.
L432-433. What is meant by status?
ok
L434-435. Is it the spatial resolution that limits the wide use of altimetry data for snow and permafrost research or the frequencies used?
Text modified.

L460+ Wouldn't it be important to mention the relevance for monitoring the Antarctic cryosphere?
Antarctica also added to the text in a sentence.

L467. Is this true? ICESat-2 reaches 88-deg latitude.
Sentence changed.

L471. It'd be helpful to restate the along-track resolution here since it is a key element

for the mission.
Ok, added to the paper.

We would like to change the reviewer for providing such a thorough list of technical changes and suggestions. We have incorporated all of these changes in the revised version of the manuscript.

**Response to Anonymous Referee #3, received and published: 26 March 2020**

**27 March 2020**

Please see our responses in blue color in the reviewers text.

Thank you for this careful review and the constructive comments below, which have been used to revise the paper.

**Anonymous Referee #2**
Overview: This paper presents the justification for and early design candidate for the CRISTAL mission. The paper also presents on the individual science applications that will be serviced by such a mission.

Summary: Firstly, apologies for my tardy review, I could would like to blame it on the new situation the world finds itself in with COVID-19 but I procrastinated well before that became a factor. A thorough presentation of the need for, and technical capabilities of, the CRISTAL mission is a welcomed contribution to the community. The mission is very exciting and will provide invaluable data necessary to advance polar sciences in the coming decades.

Overall, I found this manuscript very difficult to review. I tried to review the manuscript multiple times but the first 4 sections were such a slog that it was unmotivating. The first 4 sections read like a cut and paste of multiple science agency white papers while the final section, section 5, provides a nice science justification for the mission but is completely decoupled from the first 4 sections. In my opinion the paper requires significant restructuring and more consideration as to what is included and why. I would start with a section (1) that provides the motivation for the mission: that is the science justification (essentially section 5) with a dedicated subsection that summarizes the EU Arctic Policy and primary user requirements that motivate the mission. (2) I would then summarize the geophysical observables that are needed to satisfy the science and application needs (maybe as a table). There is a table for radar specification but not for the geophysical observation requirements that are more relevant to an overview paper. This could be followed by a section (3) that describes the CRISTAL mission capabilities showing that it meets the needs presented in the prior section. The current section 2 could be reduced to 1 or 2 paragraphs that get included in the new section (3), no need to review the full Copernicus program. I think nearly all of the material is in the document, it just needs to be reworked into a coherent justification and description of the mission.

Thank you for this important comment. We have restructured the paper to address this comment. It should be noted, however, that addressing Copernicus services requirements (operational) in the context of the High-Priority Candidate Missions remain the predominant justification for the mission. The science needs are undoubtedly important.
To take into account the reviewers comment, we have re-organised the paper by changing the title, the abstract, shortened section 2 (into 3 paragraphs) and interchanged sections 4 and 5 after modifying them to reduce repetition.

I have many small comments on grammar and wording but I don't think the manuscript is at the point yet where those details would be all that useful. For that reason, I only list my major comments here:
- There is a reasonable amount of redundant text between section 5 and the rest of the manuscript. This is mostly a result of organization and can be fixed by structuring and being more selective on what's included

We have restructured the paper to remove repeating sentences and redundant text as much as possible.

- Sections 1-4 lack references supporting statements (I believe there is only 1 reference supporting the idea that the combination of ka- and ku- band altimeter will substantially improve measurements of snow on sea ice.

More references have been added. It is true that more experimental data is required further study the combination of Ka-band and Ku-band and its impact.

-Also several citations are outdated (e.g. Chen 2013), DeCanto and Pollard, 2016 is highly controversial and is not a representative citation for a current best estimate of future ice sheet change.

This has been changed and replaced with more representative citations.

- There is virtually no justification for including a microwave radiometer and no supporting literature, this should be shored up with a clear justification as to the applications (with list of geophysical observables) and science it will support with appropriate citations

Agreed. A justification is now added to the paper. There are perhaps two points to mention:
- Over sea ice, the active/passive synergy enables to classify the sea ice type
- Over open ocean, the radiometer is required to meet the range accuracy requirement.

- Observing mode "Land-ice and Glacier" doesn't make a lot of sense since glaciers are Land-ice. I would suggest renaming this mode to "Land-ice" -or- glaciers and permafrost (GP) –or glaciers, ice sheets and permafrost (GISP)

Understood. However, the names for the observing modes have now been used throughout the mission preparation phase, with different study partners and industry. We would prefer to keep these names in order not to confuse partners involved. Acronyms have been removed for better clarity.

- It is not clear how CRYSTAL will handle range ambiguities over complex glacier and permafrost terrain as advertised.

The purpose of the interferometric mode is to allow the across-track angle offset of the echoing point to be determined directly (using interferometric phase information). Phase-wrapping can occur when the across track offset is great enough and this can have the effect of the echo appearing to come from the other side of the ground track. This restricts the method to areas with an average cross-track slope of ~ 0.5 to ~ 2◦. Normally, this is flagged by use of an ambiguity Digital Elevation Model.

A sentence is added.

- The colors for Figure 3 are difficult to distinguish (particularly magenta and purple). The figure should also include a colorbar and pole hole data gap.

Figure 3 shows the required masks for operation, which is the reason why the polar gap is not relevant here. The figure caption was updated and improved.

[revised manuscript text omitted]

---

## Referee Report (RR1)

[referee-annotated manuscript omitted]

---

## Author Response (AR2)

[revised manuscript text omitted]

**Response to Referee #1, submitted on 14 May 2020**

**12 June 2020**

Suggestions for revision or reasons for rejection (will be published if the paper is accepted for final publication)
I thank the authors for thoroughly addressing the concerns and specific comments raised by each reviewer. The paper is hugely improved and I enjoyed reading it! To summarise:
- The change in title and abstract is great make the scope of the paper clear
- Addition of Tables 1 and 2 provide a large amount of information in a digestible format
- The paper flows well with improved cogerence between sections
- Timeframe of mission development is clear

A couple of (very small) comments that should be considered before publication:
- P4L117: "Earlier, the Global Climate Observing System (GCOS, 2011) pointed out that…" to "The Global Climate Observing System (GCOS, 2011) have stated that…"
- The authors note that Section 5.1 now contains almost 25 references, Section 5.3 8 references. I understand that the paper cannot provide a complete overview of all aspects covered in the literature, but I think all those included now are necessary. If references need to be reduced on request of the editor, please make sure you have a balanced representation of institutions, which was not the case in the original submission.

Please see our responses in blue color below to the editorials and comments in the paper.

We thank the referee for the comments and suggestions. The paper has been updated taking into account the comments made. The number of references have not been reduced.

**Response to Referee #3, submitted on 28 May 2020**

**12 June 2020**

Suggestions for revision or reasons for rejection (will be published if the paper is accepted for final publication)

The authors have done a commendable job restructuring the paper and it now reads much better than the original submission. For efficiency, I have provided comments and suggested edits directly on the pdf version of the manuscript. Overall, I think the paper is in pretty good shape and provides valuable information on the motivation and concept design for the proposed CRYSTAL mission that, if launched, will play a critical role in monitoring environmental change of polar regions in the coming decade and will help advance our understanding of the interconnected nature of polar and global climate.
Referee Report: tc-2020-3-referee-report.pdf

Please see our responses in blue color below to the editorials and comments in the paper. The Line numbers refer to the line numbers in the referee report.

We thank the referee for the detailed corrections and suggestions. We have performed most of the suggested changes in the text. We comment on the explicit comments in the pdf or where we have not changed the text.

Line 29. Acronyms in Abstract all removed.

Lines 51ff. We have kept the paragraph as is. Such introductory paragraph is necessary in our view to set the stage for the mission.

Line 83. Paragraphs linked.

Line 145ff. Table 1. We have changed the table for the suggested formatting edits. However, we have not changed the wording of the Land ice mission objectives.

Line 162. Added space.

Line 199. --> Response to review comment:

"There should be an explicit mention of how CRISTAL will compliment ICESat-2 that has improved accuracy over what is reported here."

As of the time of writing, the uncertainty of ICESat-2 sea ice thickness estimates has not yet been evaluated through validation.

However we added a statement regarding ICESat sea ice thickness uncertainty, which is similar to that of CryoSat-2. We have also added a reference to ICESat-2.

Line 224. References added.

Line 244. References added.

Line 252. References not added here. The suggested 'Smith' reference is added at another place, see below.

Line 266. Sentences added to clarify the uncertainty requirement.

Line 300. Suggested Smith et al reference added.

Line 322. Sentence modified and references added. The radiometer on-board of CRISTAL will add value to meet the range accuracy requirement of a wet tropospheric correction with respect to a model (Picard et al., 2015, Legeais et al., 2014, and Vieira et al., 2019).

Line 343. Sentence kept.

Line 413. 'Land ice and Glacier mode' changed to 'Land ice mode' to be also consistent with Table 1, 3 and Figure 3.

References reviewed and updated.

---

## Author Response (AR3)

[revised manuscript text omitted]

**Response to Editor, submitted on 18 June 2020**

**18 June 2020**

Dear Michael - Thanks very much for the revised manuscript. You have clearly addressed the review comments and the manuscript is now accepted for publication in The Cryosphere. I have just a few minor issues to address:
Line 160-162: Can you update the sea ice trends information through 2019? I believe the sea ice extent trend is virtually unchanged at -13% per decade (check the NSIDC sea ice analysis page). The proportional loss of MYI was recently updated as part of the IPCC SROCC.
Line 176: perhaps cite the IPCC SROCC polar regions chapter (or references therein) here to support the statement on shipping trends?
Line 318: not clear what is meant by "with respect to a model".
Line 355: "Measurements as provided by CRISTAL may, however, be useful in retrieving internal properties of the snowpack." Can a citation be provided to provide support to this statement?

Thanks very much for your contribution to The Cryosphere and please proceed with submission of the final manuscript version.
Chris Derksen

Please see our responses in blue color below to the editorials and comments in the paper. The Line numbers refer to the line numbers in the referee report.

We thank the editor for the corrections and suggestions. We have performed the suggested changes in the text.

Line 160-162: Can you update the sea ice trends information through 2019? I believe the sea ice extent trend is virtually unchanged at -13% per decade (check the NSIDC sea ice analysis page). The proportional loss of MYI was recently updated as part of the IPCC SROCC.

Ok, changed.

Line 176: perhaps cite the IPCC SROCC polar regions chapter (or references therein) here to support the statement on shipping trends?

Ok, cited.

Line 318: not clear what is meant by "with respect to a model".

Sentence slight adapted to that it becomes clearer.

Line 355: "Measurements as provided by CRISTAL may, however, be useful in retrieving internal properties of the snowpack." Can a citation be provided to provide support to this statement?

Sentence adapted and new references added.